# Sample-wise Adaptive Weighting for Transfer Consistency in Adversarial Distillation

**Hongsin Lee**                                                    *hongsin04@kaist.ac.kr*
*School of Electrical Engineering*
*Korea Advanced Institute of Science and Technology (KAIST)*

**Hye Won Chung**                                                  *hwchung@kaist.ac.kr*
*School of Electrical Engineering*
*Korea Advanced Institute of Science and Technology (KAIST)*

**Reviewed on OpenReview:** *https://openreview.net/forum?id=ek45VamPCE*

## Abstract

Adversarial distillation in the standard min–max adversarial training framework aims to transfer adversarial robustness from a large, robust teacher network to a compact student. However, existing work often neglects to incorporate state-of-the-art robust teachers. Through extensive analysis, we find that stronger teachers do not necessarily yield more robust students–a phenomenon known as robust saturation. While typically attributed to capacity gaps, we show that such explanations are incomplete. Instead, we identify adversarial transferability–the fraction of student-crafted adversarial examples that remain effective against the teacher–as a key factor in successful robustness transfer. Based on this insight, we propose Sample-wise Adaptive Adversarial Distillation (SAAD), which reweights training examples by their measured transferability without incurring additional computational cost. Experiments on CIFAR-10, CIFAR-100, and Tiny-ImageNet show that SAAD consistently improves AutoAttack robustness over prior methods. The code is available at `https://github.com/HongsinLee/saad`.

## 1 Introduction

Deep neural networks have achieved remarkable success across diverse domains, yet they remain highly susceptible to adversarial perturbations (Goodfellow et al., 2014; Carlini & Wagner, 2017; Madry et al., 2017; Athalye et al., 2018), posing significant risks in safety-critical applications (Grigorescu et al., 2020; Ma et al., 2021; Wang et al., 2023a). In response, a variety of defense strategies have been proposed (Das et al., 2017; Cohen et al., 2019; Carmon et al., 2019; Xie et al., 2019; Zhang et al., 2022; Jin et al., 2023), among which adversarial training (AT) (Goodfellow et al., 2014; Madry et al., 2017) has emerged as a leading method. Despite its effectiveness, AT typically requires large-scale models, resulting in a substantial performance gap for lightweight architectures commonly deployed in resource-constrained settings (Madry et al., 2017). To bridge this gap, AT-based *adversarial distillation (AD)* methods (Goldblum et al., 2020; Zhu et al., 2021; Zi et al., 2021; Maroto et al., 2022; Huang et al., 2023; Jung et al., 2024; Park & Min, 2024; Lee et al., 2025) have been proposed as a promising approach for transferring the robustness of large teacher models to compact student models.

Despite the promise of AD, many existing studies often neglect to incorporate state-of-the-art robust teachers from standardized benchmarks such as RobustBench (Croce et al., 2021). A natural expectation is that a more robust teacher would yield a correspondingly robust student. However, as illustrated in Figure 1a, our experiments reveal that employing stronger teachers can in fact degrade student robustness, contradicting conventional intuition. This surprising result raises a fundamental question: what factors account for the variability in AD performance across teacher models, and why does greater teacher robustness not necessarily

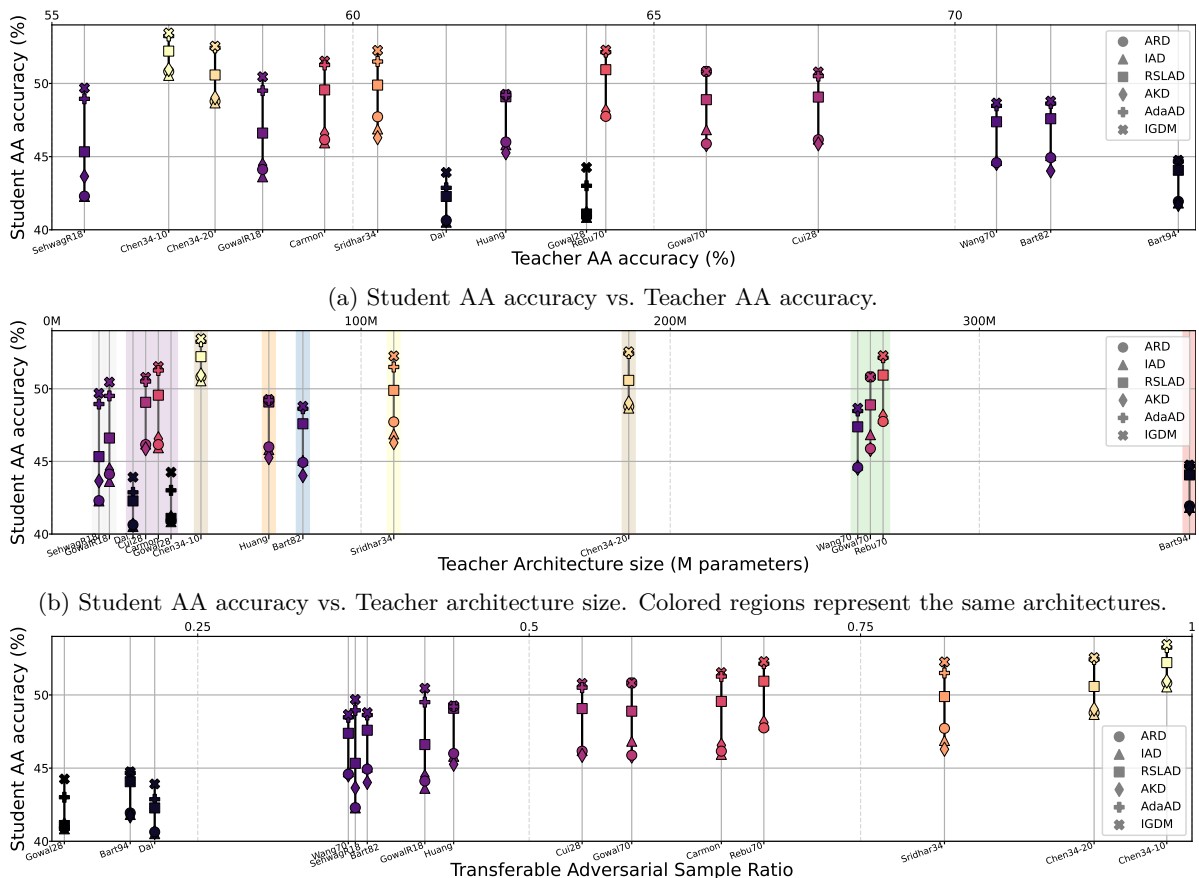

(a) Student AA accuracy vs. Teacher AA accuracy.

(b) Student AA accuracy vs. Teacher architecture size. Colored regions represent the same architectures.

(c) Student AA accuracy vs. Fraction of adversarial examples that transfer from student to teacher (TAS).

Figure 1: Adversarial distillation results on CIFAR-10 with a ResNet-18 student. Points are color-coded by the TAS ratio to illustrate the consistent correlation between adversarial transferability and resulting student robustness. Detailed teacher information and full experimental results are provided in Section A.1.

lead to more effective robustness transfer? In this work, we address this question by examining the role of *adversarial transferability* in robust knowledge distillation.

Previous studies have attributed the failure of robustness transfer in AD to the *robust saturation effect* (Zi et al., 2021), which posits that beyond a certain capacity threshold, further increases in teacher robustness or model size yield diminishing returns for the student. However, as shown in Figure 1b, even when teachers are ordered by architectural size within the same model family (e.g., WRN-28-10 in purple and WRN-70-16 in green), student robustness varies significantly. This suggests that capacity gap alone cannot fully explain the observed discrepancies.

To better understand this limitation, we analyze how the teacher's output confidence on student-crafted attacks influences the student's adversarial variance and overfitting through distillation. We find that highly confident (i.e., low-entropy) outputs from robust teachers exacerbate student variance under attack, resulting in unstable training. Our analysis further reveals that this instability arises from a lack of *transferable adversarial samples (TAS)*–student-generated adversarial inputs that remain effective against the teacher– whose abundance strongly correlates with successful robustness transfer, as demonstrated in Figure 1c.

Motivated by this insight, we propose *Sample-wise Adaptive Adversarial Distillation (SAAD)*, a novel approach that emphasizes samples with high adversarial transferability to improve robustness transfer. SAAD assigns lower weights to non-transferable samples, effectively mitigating their high-variance effects and improving the student model's robustness. We further introduce a clean distillation term weighted by inverse

transferability, offering a tunable trade-off to recover clean accuracy without severely compromising robustness. Extensive experiments demonstrate that our method consistently improves student robustness in cases where superior teacher models did not translate into enhanced robustness under existing methods. Our contributions are as follows:

- We identify adversarial transferability as a key factor for effective adversarial distillation, explaining why stronger teachers can fail to improve student robustness.

- We propose *Sample-wise Adaptive Adversarial Distillation (SAAD)*, which selectively emphasizes transferable samples to mitigate high-variance effects and improve robustness.

- We show that our method consistently improves adversarial robustness and provides a tunable clean-robustness trade-off, outperforming prior distillation approaches.

## 2 Related Works

**Adversarial Attacks and Transferability.** Based on the adversary's level of access to the victim model, adversarial attacks are distinguished as either white-box or black-box. In the white-box paradigm, the adversary has full access to the model parameters and gradients, which enables gradient-based attacks such as FGSM (Goodfellow et al., 2014), PGD (Madry et al., 2017), and stronger optimization-based methods (Carlini & Wagner, 2017; Croce & Hein, 2020; Li et al., 2024). In contrast, black-box attacks operate with limited knowledge of the target and are typically either query-based or transfer-based. Query-based attacks directly probe the model, including score-based methods (Uesato et al., 2018; Andriushchenko et al., 2020) and decision-based boundary attacks (Chen & Gu, 2020; Chen et al., 2020; 2021b). Transfer-based attacks rely on the adversarial transferability phenomenon, where adversarial examples created for one model succeed in misleading another (Szegedy et al., 2013; Papernot et al., 2016; 2017; Tramèr et al., 2017). In this context, the adversary typically constructs adversarial examples on surrogate models and leverages their transferability as an attack mechanism (Liu et al., 2016; Mahmood et al., 2021). Diverging from this conventional paradigm, our work re-purposes adversarial transferability as a diagnostic tool. We leverage it not to attack models, but to evaluate the efficacy of different teacher models within the adversarial distillation framework.

**Adversarial Training.** In response to adversarial attacks, adversarial training (AT) has emerged as one of the most effective defenses. In its standard form, known as PGD-AT (Madry et al., 2017), the model parameters $\boldsymbol{\theta}$ are optimized via a min-max formulation:

$$\arg\min_{\boldsymbol{\theta}} \ \mathbb{E}_{(\mathbf{x},y)\sim\mathcal{D}}\Big[\mathrm{CE}(\mathbf{y}, f_{\boldsymbol{\theta}}(\mathbf{x}+\boldsymbol{\delta}))\Big], \quad \text{where} \quad \boldsymbol{\delta} = \arg\max_{\boldsymbol{\delta}\in\Delta} \mathrm{CE}(\mathbf{y}, f_{\boldsymbol{\theta}}(\mathbf{x}+\boldsymbol{\delta})) \tag{1}$$

Here, the inner maximization generates adversarial perturbations that maximize the cross-entropy loss, while the outer minimization trains the model to minimize this loss under the worst-case perturbation $\boldsymbol{\delta}$. To address trade-offs between robustness and accuracy, TRADES (Zhang et al., 2019) reformulates adversarial training by decoupling the loss into a clean classification term and a robustness regularization via KL divergence:

$$\arg\min_{\boldsymbol{\theta}} \ \mathbb{E}_{(\mathbf{x},y)\sim\mathcal{D}} \left[\mathrm{CE}(\mathbf{y}, f_{\boldsymbol{\theta}}(\mathbf{x})) + \lambda \cdot \max_{\boldsymbol{\delta}\in\Delta} \mathrm{KL}(f_{\boldsymbol{\theta}}(\mathbf{x})\|f_{\boldsymbol{\theta}}(\mathbf{x}+\boldsymbol{\delta}))\right] \tag{2}$$

This formulation explicitly balances natural accuracy and robustness through the hyperparameter $\lambda$. MART (Wang et al., 2020) integrates per-sample weighting based on prediction confidence. Its objective can be described as:

$$\arg\min_{\boldsymbol{\theta}} \ \mathbb{E}_{(\mathbf{x},y)\sim\mathcal{D}}\left[\mathrm{CE}(\mathbf{y}, f_{\boldsymbol{\theta}}(\mathbf{x}+\boldsymbol{\delta})) + (1-w_y) \cdot \mathrm{KL}(f_{\boldsymbol{\theta}}(\mathbf{x})\|f_{\boldsymbol{\theta}}(\mathbf{x}+\boldsymbol{\delta}))\right] \tag{3}$$

where inner maximization to compute $\boldsymbol{\delta}$ is equal to the PGD-AT and the weight $w_y$ is computed from the confidence of the true class prediction. This adaptively emphasizes hard examples and misclassified inputs during training. These variants have inspired a rich line of adversarial training research (Qin et al., 2019; Wu et al., 2020; Bai et al., 2021; Jin et al., 2022; Tack et al., 2022; Jin et al., 2023; Wei et al., 2023).

**Adversarial Distillation.** Adversarial distillation (AD) aims to transfer the robustness of a large, adversarially trained teacher model into a more compact student model. The dominant paradigm, and the focus of our work, is AT-based AD, which leverages teacher signals under a min-max adversarial training framework (Goldblum et al., 2020; Zhu et al., 2021; Zi et al., 2021; Maroto et al., 2022; Huang et al., 2023; Kuang et al., 2023; Lee et al., 2025). Unlike standard knowledge distillation (Hinton et al., 2015), which aligns clean predictions, this approach explicitly considers adversarially perturbed inputs during training to preserve robustness in the student.

Adversarial Robustness Distillation (ARD) (Goldblum et al., 2020) initiates this line of work by incorporating adversarial examples into the distillation process, showing that robust teachers can effectively guide student models when both are trained under adversarial settings. RSLAD (Zi et al., 2021) builds on this by integrating teacher outputs directly into the generation of adversarial examples, encouraging smoother teacher logits and more stable student learning, and further reports a *robust saturation effect*: a student's robustness increases with teacher strength only up to a moderately larger teacher and then declines as teacher capacity outpaces the student. Introspective Adversarial Distillation (IAD) (Zhu et al., 2021) proposes a confidence-based modulation of the teacher signal, weighting the distillation loss by the estimated reliability of the teacher under adversarial inputs. AdaAD (Huang et al., 2023) introduces a more sophisticated approach where the teacher is actively involved in the inner maximization step, generating adversarial examples that are optimized with respect to both the student and the teacher. Most recently, IGDM (Lee et al., 2025) indirectly distills the gradient information of the teacher model to enhance the robustness further. Table 11 summarizes the inner maximization and outer minimization objectives used by representative AD methods.

While the aforementioned methods distill from a single robust teacher, another line of research employs multiple teachers to address the trade-off between clean accuracy and robustness (Zhao et al., 2022; Deng et al., 2024). AD has also been applied to broader robustness contexts such as class imbalance (Yue et al., 2023; Zhao et al., 2024; Cho et al., 2025b), incremental learning (Cho et al., 2025a), and self-distillation (Jung et al., 2024). Another line of research transfers robustness using non-AT-based methods. These approaches often leverage gradient or feature matching on clean inputs (Shafahi et al., 2019; Chan et al., 2020; Awais et al., 2021; Chen et al., 2021a; Muhammad et al., 2021; Shao et al., 2021; Vaishnavi et al., 2022). As these methods are designed to replace the expensive PGD inner-loop, they optimize for a different trade-off and inherently sacrifice robustness. They are therefore orthogonal to our work, which focuses on diagnosing and solving the robust saturation phenomenon within the AT-based paradigm.

## 3 Robust Teacher Failures: Entropy, Variance, Transferability

In prior AD works, the teacher models are typically either large networks trained with methods such as TRADES (Zhang et al., 2019) or publicly available robust models widely adopted by early AD research (Zi et al., 2021; Huang et al., 2023; Lee et al., 2025). Although a new generation of SOTA robust models are now readily available on RobustBench (Croce et al., 2021), many recent AD studies have not focused on incorporating these specific models. One might naturally expect that leveraging stronger teachers would yield improved student robustness. However, our experiments across a diverse set of teachers in Figure 1 reveal that existing AD methods are highly susceptible to teacher choice, with even the most robust teachers leading to poor student robustness.

A simple explanation often given for this phenomenon is the so-called robust saturation effect (Zi et al., 2021), which attributes the diminishing gain of adversarial distillation to the capacity gap between the teacher and student models. However, as shown in Figure 1b, we find no consistent trend even when distillation outcomes are sorted by teacher architecture, indicating that the capacity gap alone cannot fully explain the failure modes. Accordingly, we introduce a new framework by dividing robust teachers into two categories: *Effective Robust Teachers* (ERTs) and *Ineffective Robust Teachers* (IRTs), defined by whether students distilled from them via recent AD methods, on average, outperform or underperform AT baselines (TRADES) in robust accuracy. To systematically compare these groups, we select representative teachers as summarized in Table 1, with additional details provided in Section A.1. For interpretability in subsequent analyses, we adopt RSLAD (Zi et al., 2021) as the baseline AD method and fix the student architecture to ResNet-18 trained on CIFAR-10.

Table 1: Comparison of distillation outcomes using different teacher models categorized as Effective Robust Teachers (ERTs) and Ineffective Robust Teachers (IRTs). AA denotes AutoAttack accuracy (%) of the teacher (left) and the student (right); RO measures robust overfitting, computed as the gap between the student's best and last PGD-20 accuracy on the test set. AVar denotes the adversarial variance, and TAS refers to the ratio of transferable samples in the training dataset.

| | Teacher Info | | | Distillation Results | | | |
|---|---|---|---|---|---|---|---|
| Group | RobustBench name | Architecture | AA | AA | RO | AVar | TAS |
| ERT | Rebuffi2021Fixing | WRN-70-16 | 64.20 | 50.94 | 0.20 | 0.0267 | 0.677 |
| | Chen2021LTD | WRN-34-10 | 56.94 | 52.21 | 0.15 | 0.0059 | 0.981 |
| IRT | Bartoldson2024Adversarial | WRN-94-16 | 73.71 | 44.07 | 5.44 | 0.0834 | 0.199 |
| | Gowal2021Improving | WRN-28-10 | 63.38 | 41.08 | 7.01 | 0.3058 | 0.149 |

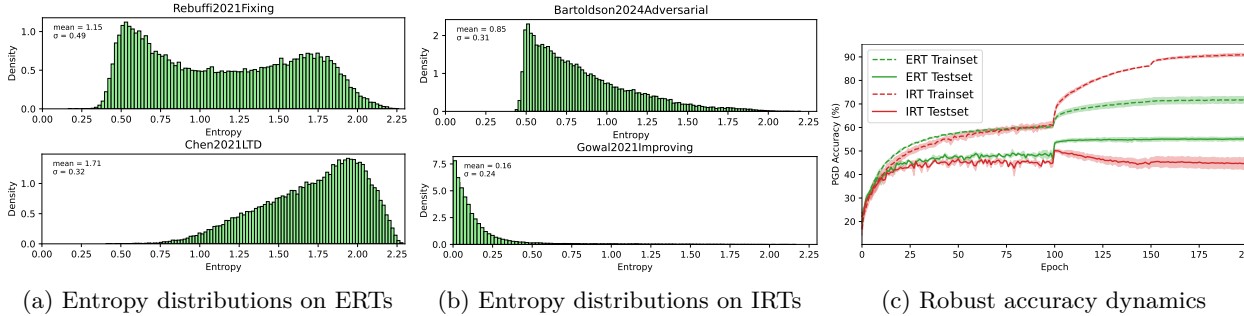

(a) Entropy distributions on ERTs     (b) Entropy distributions on IRTs     (c) Robust accuracy dynamics

Figure 2: **(a)** Density histograms of teacher-logit entropies on student-generated PGD-20 adversarial training inputs for two ERTs. **(b)** Same, but for IRTs. **(c)** PGD-20 robust accuracy on training and test sets across epochs for students distilled from individual teachers within the ERT and IRT groups. Solid lines indicate group-wise averages, and shaded regions represent standard deviations across teachers in each group.

## 3.1 Characterizing IRTs: Overconfidence and Overfitting

We observe two key distinctions between IRTs and ERTs. First, IRTs tend to produce lower-entropy outputs than ERTs, particularly on adversarial inputs generated by the student model. Figure 2a and Figure 2b show the density histograms of teacher-logit entropies evaluated on student-generated PGD-20 adversarial inputs. We find that IRTs yield highly confident predictions with significantly lower entropy, while ERTs maintain a broader entropy distribution, suggesting a more calibrated uncertainty. Importantly, a high output entropy does not necessarily imply non-robustness of the teacher model. Despite exhibiting higher entropy, ERTs can correctly classify adversarial examples crafted on student models. This suggests that ERTs maintain a level of uncertainty around adversarial inputs without fully collapsing into overconfident predictions, whereas IRTs often yield overconfident outputs aligned closely with the true label, even under attack.

Second, students distilled from IRTs exhibit pronounced robust overfitting, whereas those distilled from ERTs maintain stable generalization. This effect is visualized in Figure 2c, where the PGD-20 robust accuracy on the training and test sets for IRTs diverges significantly after the learning rate decay—a characteristic pattern of robust overfitting driven by the disruption of the min–max balance caused by the decay (Wang et al., 2023b). To quantify this, we report robust overfitting (RO) as the gap between the student's best and last PGD-20 accuracy on the test set in Table 1; IRT-distilled students exhibit large RO values, while ERT students show minimal overfitting. These results demonstrate a clear empirical link between overconfident teacher outputs and robust overfitting in the student. While the mechanism behind this link remains unclear, the consistency of these patterns across multiple IRTs suggests a deeper connection. In the next section, we formally investigate this connection by analyzing adversarial variance as a potential explanatory factor.

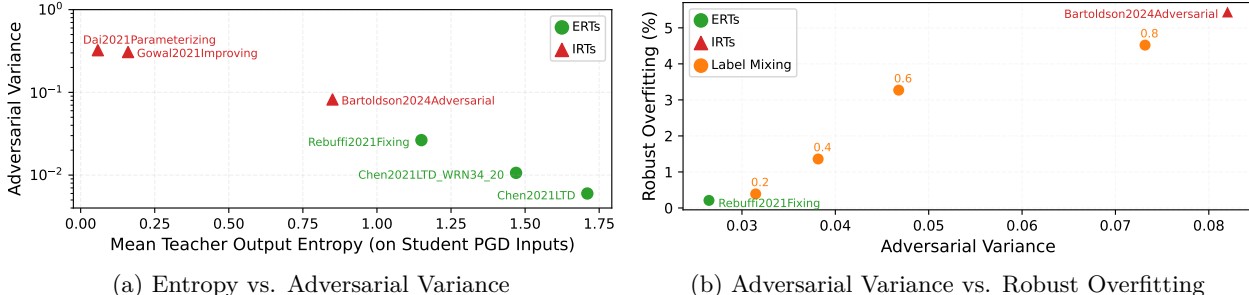

(a) Entropy vs. Adversarial Variance  (b) Adversarial Variance vs. Robust Overfitting

Figure 3: (a) Teachers with lower entropy on student-generated PGD inputs induce higher adversarial variance in the student. (b) Higher adversarial variance is associated with increased robust overfitting. Orange points show experiments where true labels are mixed into `Rebuffi2021Fixing` outputs. The numeric labels indicate the proportion of true label supervision.

## 3.2 Adversarial Variance Analysis on Adversarial Distillation

To investigate how overconfident soft labels from robust teachers induce robust overfitting in students, we extend the classical bias–variance decomposition of expected risk to the adversarial distillation setting by introducing adversarial variance. This formulation unifies earlier decompositions from adversarial training (Yu et al., 2021) and knowledge distillation (Zhou et al., 2021), and helps account for teacher-dependent variation in adversarial distillation. We note that $f_{\hat{\boldsymbol{\theta}}(\mathcal{D})} : \mathcal{X} \to \mathbb{R}^C$ represents the student model adversarially trained on dataset $\mathcal{D}$ under soft-label supervision from a fixed teacher. While the teacher remains unchanged during training, its output distribution governs the soft labels used in the distillation process, thus indirectly shaping $\hat{\boldsymbol{\theta}}$ and the student's final behavior. For a test sample $\mathbf{x}$ with ground truth label $\mathbf{y} = t(\mathbf{x}) \in \mathbb{R}^C$, we consider the worst-case perturbation

$$\boldsymbol{\delta}(\mathbf{x}, \mathbf{y}, \mathcal{D}) \in \arg\max_{\boldsymbol{\delta} \in \Delta} L_{\max}\big(f_{\hat{\boldsymbol{\theta}}(\mathcal{D})}(\mathbf{x} + \boldsymbol{\delta}), \mathbf{y}\big), \tag{4}$$

and define

$$\hat{\mathbf{y}} := f_{\hat{\boldsymbol{\theta}}(\mathcal{D})}(\mathbf{x} + \boldsymbol{\delta}(\mathbf{x}, \mathbf{y}, \mathcal{D})), \qquad \bar{\mathbf{y}} := \frac{1}{Z} \exp\left(\mathbb{E}_{\mathcal{D}}[\log \hat{\mathbf{y}}]\right), \tag{5}$$

where $\bar{\mathbf{y}}$ denotes the normalized geometric mean of student predictions $\hat{\mathbf{y}}$ over datasets $\mathcal{D}$, and $Z$ is the normalization constant to ensure $\bar{\mathbf{y}}$ lies in the probability simplex. Then, the expected adversarial cross-entropy risk admits the following decomposition:

$$\text{ARisk} = \mathbb{E}_{\mathbf{x}, \mathcal{D}}\left[\text{CE}(\mathbf{y}, \hat{\mathbf{y}})\right] = \underbrace{\mathbb{E}_{\mathbf{x}}\left[-\mathbf{y}\log\mathbf{y}\right]}_{\text{Intrinsic Noise}} + \underbrace{\mathbb{E}_{\mathbf{x}}\left[\mathbf{y}\log\frac{\mathbf{y}}{\bar{\mathbf{y}}}\right]}_{\text{Adversarial Bias}} + \underbrace{\mathbb{E}_{\mathbf{x}, \mathcal{D}}\left[\text{KL}(\bar{\mathbf{y}} \,\|\, \hat{\mathbf{y}})\right]}_{\text{Adversarial Variance}}, \tag{6}$$

where $\text{CE}(\mathbf{p}, \mathbf{q}) = -\sum_i p_i \log q_i$ is the cross-entropy loss. Detailed explanation and an algorithm for estimating the adversarial bias and variance are given in Section A.2. This decomposition enables us to empirically analyze how the adversarial variance of student models in adversarial distillation varies depending on different teacher models.

**Overconfident Soft Labels Induce High Adversarial Variance.** We observe that adversarial variance increases when the teacher produces low-entropy predictions on student-generated PGD inputs. As shown in Figure 3a, robust teachers such as `Gowal2021Improving` and `Bartoldson2024Adversarial`—despite their high standalone robustness—exhibit low entropy and correspondingly high student variance, whereas higher-entropy teachers such as `Rebuffi2021Fixing` and `Chen2021LTD` yield more stable behavior. This suggests that overconfident teacher outputs undermine the regularizing effect of soft labels, amplifying variance during adversarial training. To probe this effect more directly, we conduct an interpolation experiment using `Rebuffi2021Fixing`, gradually injecting the ground-truth label into the teacher's logits. As illustrated in Figure 3b, adversarial variance increases monotonically with the interpolation coefficient, indicating that growing confidence in soft labels directly induces instability in the student's response.

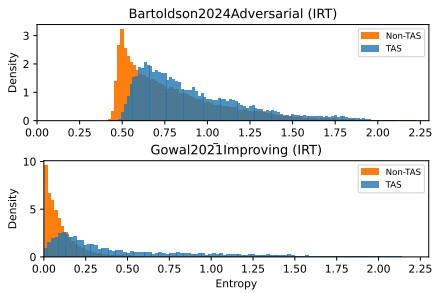
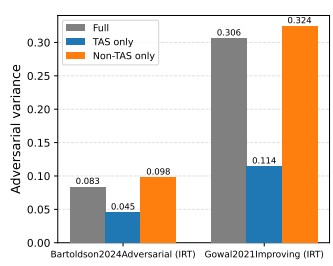
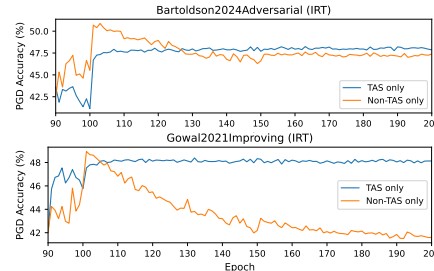

(a) TAS and Non-TAS group entropy on IRTs

(b) Variance on TAS and Non-TAS Subsets on IRTs

(c) Test Robust Accuracy Trained on TAS and Non-TAS Subsets on IRTs

Figure 4: **(a)** Density histograms of teacher-logit entropies on student-generated PGD-20 adversarial training input, separated into TAS and Non-TAS groups. **(b)** Bar chart of adversarial variance when training only on the TAS subset versus the Non-TAS subset (for each teacher). **(c)** PGD-20 robust accuracy on train (dashed) and test (solid) over epochs.

**High Adversarial Variance Causes Robust Overfitting.** Analogous to classical statistical learning theory, we find that adversarial variance, measured over perturbed inputs, serves as a strong indicator of robust overfitting. As shown in Figure 3b, we observe a clear correlation between the magnitude of adversarial variance and the degree of overfitting to adversarial training data. While AD is generally expected to reduce variance and thereby mitigate overfitting, we find that this effect depends critically on the teacher's output distribution. In earlier AD studies, robust overfitting received limited attention, likely because ERTs inherently produce soft labels with sufficient uncertainty, resulting in low adversarial variance. However, as more powerful yet sharper teachers are adopted, understanding and controlling adversarial variance becomes essential for ensuring stability in robust distillation.

Overconfident soft labels from IRTs induce high adversarial variance in the student model, leading to robust overfitting. While this explains the mechanism of failure, it raises a deeper question: why do IRTs fail to provide meaningful supervision on student-generated adversarial examples? In the following section, we show that this limitation arises from a lack of transferability between the adversarial behaviors of the student and teacher models.

### 3.3 Sample-Level Transferability in Adversarial Distillation

We identify the lack of *transferable adversarial samples* (TAS) as the primary cause of failure in AD under IRT supervision. Specifically, when a student-crafted perturbation fails to induce a comparable adversarial shift in the teacher's prediction, the teacher's supervision signal becomes misaligned, thereby degrading the efficacy of robustness transfer. To formalize this notion, we conduct a sample-level analysis of behavioral alignment between student and teacher models under adversarial perturbations. We define a transferable adversarial sample as an input $\mathbf{x}$ for which the adversarial perturbation $\boldsymbol{\delta}_S$, crafted by the student, induces a response from the teacher that aligns more closely with its own adversarial response than with its original (clean) prediction. Formally, this condition is satisfied if:

$$\mathrm{KL}\big(f_T(\mathbf{x} + \boldsymbol{\delta}_S) \,\|\, f_T(\mathbf{x})\big) \geq \mathrm{KL}\big(f_T(\mathbf{x} + \boldsymbol{\delta}_S) \,\|\, f_T(\mathbf{x} + \boldsymbol{\delta}_T)\big), \tag{7}$$

indicating that the student's adversarial perturbation is aligned well with the teacher's $\boldsymbol{\delta}_T$.

In Figure 4a, we compare the output entropy of IRT teachers on student-generated adversarial inputs, separating samples into TAS vs. non-TAS categories. We observe that the TAS group maintains higher entropy, while the non-TAS group is concentrated in the low-entropy regime, indicating overconfident predictions. Furthermore, Figure 4b presents the adversarial variance observed when training is continued separately on each sample group following 90 epochs of warm-up on the full dataset. Non-TAS group results in significantly higher adversarial variance, suggesting that the supervision they provide is unstable due to misaligned adversarial behavior. Further, Figure 4c shows that this instability correlates with degraded generalization:

models trained on the non-TAS group exhibit pronounced robust overfitting, whereas training on the TAS group preserves robust generalization. Taken together, these findings indicate that non-TAS, characterized by low entropy and high adversarial variance, induce unstable and misaligned supervision, ultimately leading to robust overfitting. Consequently, a high proportion of non-transferable examples impairs the efficacy of adversarial distillation, indicating the importance of TAS for successful robustness transfer.

As shown in Table 1, the proportion of TAS is substantially lower for IRTs compared to ERTs, further reinforcing the connection between transferability and successful robustness transfer. Moreover, Figure 1c demonstrates a positive correlation between the TAS ratio and the student model's robustness under AutoAttack, underscoring the predictive value of this metric. These observations suggest that the scarcity of TAS is not merely a byproduct of poor distillation but a central cause of IRTs' inability to provide effective supervision. Thus, sample-level transferability emerges as a critical factor in explaining and potentially overcoming the limitations of adversarial distillation.

## 4 Main Method: Sample-wise Adaptive Adversarial Distillation

Our analysis reveals that the difference in distillation effectiveness between ERTs and IRTs arises from the entropy distribution of teacher logits and the resulting adversarial variance. ERTs produce higher-entropy outputs on student-generated adversarial inputs, which lead to lower adversarial variance and better generalization. In contrast, IRTs yield overconfident, low-entropy predictions that induce high adversarial variance and robust overfitting. This problem is further pronounced by the large portion of non-TAS samples under IRTs, where adversarial perturbations fail to meaningfully alter the teacher's outputs. These non-TAS samples dominate training, thereby exacerbating variance-driven overfitting.

While existing AD methods can be effective when the teacher provides a sufficient number of transferable samples, they apply the distillation objective uniformly across all data points, failing to distinguish between transferable and non-transferable samples. A simple alternative is to train only on transferable samples. However, as shown in Table 2, this approach yields inferior overall performance due to the reduced sample count, despite improved robustness over training only on non-transferable samples. These findings suggest that entirely discarding non-transferable samples is suboptimal, especially as adversarial training demands intensive data to achieve robustness (Schmidt et al., 2018).

Table 2: Impact of transferable adversarial samples on adversarial distillation.

| Setting | # of Data | Clean | AA |
|---|---|---|---|
| Full Data | 50000 | 84.28 | 44.42 |
| Excluding TAS | 45161 | 83.93 | 43.05 |
| Only on TAS | 4839 | 80.70 | 44.00 |

Motivated by these insights, we propose Sample-wise Adaptive Adversarial Distillation (SAAD), which assigns higher weights to transferable adversarial samples during distillation. The weighting mechanism is derived from the transferable adversarial sample criterion defined in equation 7. A key challenge, however, is that computing the teacher-side perturbation $\boldsymbol{\delta}_T$, which is not required in standard adversarial distillation, incurs additional computational overhead, particularly for large teacher models. To address this, we note that from the student's perspective, the teacher outputs $f_T(\mathbf{x})$ and $f_T(\mathbf{x} + \boldsymbol{\delta}_T)$ remain fixed, while only the student-induced perturbation $\boldsymbol{\delta}_S$ varies. We further leverage the empirical observation that the teacher's output distribution on its own adversarial input, $f_T(\mathbf{x} + \boldsymbol{\delta}_T)$, typically exhibits higher entropy than on the clean input $f_T(\mathbf{x})$. According to equation 7, transferable samples are those for which the student's perturbation $\boldsymbol{\delta}_S$ sufficiently approximates the teacher's own adversarial behavior, thereby inducing a comparable increase in entropy in $f_T(\mathbf{x} + \boldsymbol{\delta}_S)$.

Based on this insight, SAAD assigns sample-wise weights proportional to the entropy of $f_T(\mathbf{x} + \boldsymbol{\delta}_S)$, effectively prioritizing transferable adversarial examples without incurring additional computational cost. A conceptual motivation for using the entropy of $f_T(x + \delta_S)$ as an empirical proxy for the TAS criterion is provided in Section A.3.1. The resulting loss function is defined as:

$$L_{\text{SAAD}} = \frac{1}{N} \sum_{i=1}^{N} w_i \cdot L_{\text{AD}}(f_S, f_T, \mathbf{x}_i, \boldsymbol{\delta}_{S,i}), \quad w_i := H\left(f_T(\mathbf{x}_i + \boldsymbol{\delta}_{S,i})\right), \tag{8}$$

Table 3: Performance (%) of the teacher models.

| Dataset | RobustBench name | Architecture | Clean | AA |
|---|---|---|---|---|
| CIFAR-10 | `Bartoldson2024Adversarial` | WRN-94-16 | 93.68 | 73.71 |
| | `Gowal2021Improving` | WRN-28-10 | 87.50 | 63.38 |
| CIFAR-100 | `Wang2023Better` | WRN-70-16 | 75.22 | 42.66 |
| Tiny-ImageNet | `Wang2023Better` | WRN-28-10 | 65.19 | 31.30 |

where $L_{\text{AD}}$ denotes an existing AD method. In our implementation, we adopt IGDM (Lee et al., 2025) as the base method; additional details are provided in Section A.3.2.

By weighting adversarial distillation according to the entropy of the teacher's perturbed outputs, non-transferable samples receive negligible weight and are effectively suppressed. Although such samples exhibit low-entropy teacher logits, indicating limited utility for robustness, they still contain confident supervision aligned with the true label. To preserve this clean signal, we introduce a complementary clean distillation loss by assigning inverse weights $1 - \tilde{w}_i$, where $\tilde{w}_i = w_i / \log C$ denotes the entropy normalized by the maximum entropy for $C$ classes:

$$L_{\text{SAAD-C}} = L_{\text{SAAD}} + \frac{1}{N} \sum_{i=1}^{N} \beta \cdot (1 - \tilde{w}_i) \cdot \text{KL}\left(f_T(\mathbf{x}_i) \,\|\, f_S(\mathbf{x}_i)\right), \tag{9}$$

for the clean distillation weight $\beta$. The second term thus reintroduces non-transferable samples into the clean distillation process, allowing the student to learn clean knowledge from the teacher. Such a clean distillation term has appeared in prior AD methods (Zi et al., 2021; Huang et al., 2023; Lee et al., 2025), but those works set their coefficient to zero in practice, as robustness losses outweighed the clean accuracy gains. In contrast, by restricting clean distillation to non-transferable samples, we achieve substantial improvements in clean accuracy with only marginal robustness degradation.

## 5 Experimental Results

### 5.1 Experiment Setup

**Adversarial Distillation Setting.** We conduct experiments on CIFAR-10, CIFAR-100 (Krizhevsky et al., 2009), and Tiny-ImageNet (Le & Yang, 2015), using standard data augmentations (random crop and horizontal flip). We compare baseline adversarial training methods—PGD-AT (Madry et al., 2017) and TRADES (Zhang et al., 2019)—with six adversarial distillation approaches: ARD (Goldblum et al., 2020), IAD (Zhu et al., 2021), RSLAD (Zi et al., 2021), AKD (Maroto et al., 2022), AdaAD (Huang et al., 2023), and IGDM (Lee et al., 2025). Further details are provided in Section B.

**Teacher and Student Models.** We employ robust teacher models summarized in Table 3, including state-of-the-art entries from RobustBench (Croce et al., 2021)[1] for CIFAR-10 and CIFAR-100. To broaden our evaluation, we also include a variant with a different architecture for CIFAR-10. All selected teachers fall under the IRT category defined in Section 3, meaning that despite strong standalone robustness, they fail to effectively transfer robustness through existing adversarial distillation. As student architectures, we use ResNet-18 (He et al., 2016a) and MobileNetV2 (Sandler et al., 2018) for CIFAR datasets and PreActResNet-18 (He et al., 2016b) for Tiny-ImageNet.

**Evaluation Setting.** We evaluate each model using five metrics: Clean, FGSM, PGD, C&W, and AutoAttack (AA) accuracy. Clean accuracy is measured on the original test set without perturbation. FGSM and PGD accuracies are obtained using adversarial examples generated by the fast gradient sign method (Goodfellow et al., 2014) and a 20-step projected gradient descent attack (Madry et al., 2017), respectively. C&W accuracy is measured under the optimization-based attack proposed in (Carlini & Wagner, 2017), while AA

---

[1] Accessed Feb 02, 2026: `https://robustbench.github.io/`

Table 4: Adversarial distillation results using two teacher and two student models on CIFAR-10. Clean, FGSM, PGD, C&W, and AA columns report accuracy (%) under each evaluation setting. Results are averaged over three random seeds.

| Model | Method | Bartoldson2024Adversarial | | | | | Gowal2021Improving | | | | |
|---|---|---|---|---|---|---|---|---|---|---|---|
| | | Clean | FGSM | PGD | C&W | AA | Clean | FGSM | PGD | C&W | AA |
| ResNet-18 | PGD-AT | 84.27 | 52.10 | 42.34 | 42.29 | 40.85 | 84.27 | 52.10 | 42.34 | 42.29 | 40.85 |
| | TRADES | 82.70 | 57.14 | 48.81 | 48.08 | 46.46 | 82.70 | 57.14 | 48.81 | 48.08 | 46.46 |
| | ARD | 84.63 | 56.57 | 44.48 | 43.44 | 41.66 | 84.39 | 52.47 | 42.18 | 42.22 | 40.79 |
| | IAD | 84.43 | 56.64 | 44.82 | 43.47 | 41.80 | 84.28 | 52.25 | 42.16 | 42.19 | 40.70 |
| | RSLAD | 84.28 | 57.20 | 47.17 | 46.07 | 44.42 | 83.83 | 51.59 | 42.03 | 42.22 | 40.57 |
| | AKD | 84.62 | 56.29 | 44.42 | 43.43 | 41.79 | 84.32 | 52.13 | 42.38 | 42.44 | 40.95 |
| | AdaAD | 85.07 | 57.54 | 47.16 | 46.06 | 44.55 | 85.04 | 53.85 | 44.65 | 44.90 | 43.27 |
| | IGDM | 84.75 | 58.38 | 47.56 | 46.43 | 44.94 | 85.67 | 58.14 | 48.58 | 46.98 | 44.76 |
| | **SAAD-C** | **85.54** | **61.92** | 53.18 | 52.05 | 50.14 | **86.39** | **60.55** | 51.91 | 52.06 | 49.72 |
| | **SAAD** | 84.27 | 61.44 | **53.39** | **52.39** | **50.34** | 83.69 | 59.74 | **52.89** | **52.36** | **50.35** |
| MobileNetV2 | PGD-AT | 83.52 | 54.92 | 44.90 | 44.29 | 41.54 | 83.52 | 54.92 | 44.90 | 44.29 | 41.54 |
| | TRADES | 81.79 | 56.50 | 49.90 | 47.54 | 46.50 | 81.79 | 56.50 | 49.90 | 47.54 | 46.50 |
| | ARD | 83.66 | 55.09 | 44.71 | 43.49 | 41.24 | 83.62 | 54.62 | 44.60 | 44.18 | 41.47 |
| | IAD | 83.85 | 55.69 | 44.98 | 43.62 | 41.35 | 83.63 | 54.81 | 44.73 | 44.19 | 41.51 |
| | RSLAD | 83.22 | 55.54 | 46.09 | 44.80 | 42.56 | 83.41 | 54.60 | 45.01 | 44.41 | 41.78 |
| | AKD | 83.60 | 55.13 | 44.31 | 43.29 | 41.03 | 83.54 | 54.79 | 44.83 | 44.19 | 41.55 |
| | AdaAD | 84.42 | 56.38 | 46.16 | 44.99 | 43.01 | 84.37 | 54.40 | 44.40 | 44.59 | 41.95 |
| | IGDM | 84.07 | 57.31 | 47.39 | 45.43 | 43.57 | 84.13 | 57.93 | 48.70 | 47.43 | 44.83 |
| | **SAAD-C** | **85.16** | **60.53** | 52.72 | 51.26 | 49.34 | **84.81** | **58.17** | 51.09 | **50.45** | 48.08 |
| | **SAAD** | 82.04 | 59.48 | **53.69** | **51.68** | **49.88** | 80.60 | 56.85 | **51.78** | 50.25 | **48.29** |

reports worst-case accuracy under the AutoAttack ensemble (Croce & Hein, 2020). All adversarial attacks are conducted under an $l_\infty$-norm constraint of 8/255.

## 5.2 Adversarial Distillation Results

Table 4 and Table 6 summarize adversarial robustness across datasets and methods. SAAD consistently achieves the best AutoAttack accuracy across all settings, outperforming conventional adversarial training as well as prior distillation techniques. Unlike existing AD methods whose performance varies significantly depending on the teacher model, SAAD maintains strong robustness even under IRT teachers. This suggests that weighting transferable samples during training is crucial for stable robustness transfer in adversarial distillation. Moreover, SAAD-C, which selectively incorporates clean supervision on non-transferable samples, provides a tunable and highly favorable trade-off to recover clean accuracy without severely compromising robustness.

We attribute the superior performance under IRTs to the mitigation of robust overfitting. As discussed in Section 3, robust overfitting arises from high adversarial variance (AVar), particularly when the teacher produces overconfident soft labels. To address this, SAAD introduces a sample-wise weighting scheme that effectively suppresses variance. As shown in Table 5, our method substantially reduces adversarial variance and dramatically mitigates robust overfitting (RO), while increasing the ratio of transferable adversarial samples (TAS). These findings confirm that lowering adversarial variance via our weighting scheme is the key factor driving the improved generalization reported in the main tables.

Table 5: Mitigation of robust overfitting and reduction of adversarial variance via sample-wise weighting with an IRT teacher.

| Method | AVar | RO | TAS |
|---|---|---|---|
| Baseline | 0.0834 | 5.44 | 0.199 |
| SAAD | 0.0385 | 0.93 | 0.326 |

Table 6: Adversarial distillation results using ResNet-18 and PreActResNet-18 as student models for CIFAR-100 and Tiny-ImageNet, respectively, with the `Wang2023Better` teacher (WRN-70-16 for CIFAR-100 and WRN-28-10 for Tiny-ImageNet). Clean, FGSM, PGD, C&W, and AA columns report accuracy (%) under each evaluation setting. Results are averaged over three random seeds.

| Method | CIFAR-100 | | | | | Tiny-ImageNet | | | | |
|---|---|---|---|---|---|---|---|---|---|---|
| | Clean | FGSM | PGD | C&W | AA | Clean | FGSM | PGD | C&W | AA |
| PGD-AT | 56.17 | 24.74 | 19.65 | 19.79 | 18.66 | 45.71 | 15.75 | 11.67 | 11.82 | 10.91 |
| TRADES | 53.37 | 28.72 | 25.12 | 23.11 | 22.32 | 42.06 | 19.58 | 17.15 | 14.03 | 13.33 |
| ARD | 58.13 | 29.71 | 24.97 | 22.22 | 20.84 | 55.69 | 30.13 | 26.62 | 22.09 | 19.90 |
| IAD | 57.59 | 29.85 | 25.21 | 22.41 | 21.00 | 53.75 | 29.85 | 26.95 | 22.40 | 20.56 |
| RSLAD | 56.68 | 30.87 | 27.27 | 23.91 | 22.66 | 53.18 | 30.14 | 27.69 | 22.86 | 21.42 |
| AKD | 58.22 | 29.14 | 24.35 | 21.80 | 20.61 | 54.48 | 27.62 | 23.59 | 19.56 | 17.73 |
| AdaAD | 58.57 | 31.72 | 28.00 | 24.40 | 23.15 | 57.26 | 31.81 | 28.80 | 23.64 | 22.11 |
| IGDM | 56.36 | 32.95 | 29.68 | 25.91 | 24.81 | 57.15 | 31.98 | 29.02 | 23.94 | 22.52 |
| **SAAD-C** | **59.57** | **36.05** | 32.52 | 28.72 | 27.21 | **57.33** | 33.06 | 29.62 | 24.16 | 22.69 |
| **SAAD** | 59.11 | 36.01 | **32.71** | **29.36** | **27.58** | 57.16 | **33.26** | **29.95** | **24.87** | **23.42** |

Table 7: Adversarial distillation results on ResNet-18 for CIFAR-10 using an ERT teacher (`Chen2021LTD_WRN34_20`).

| Method | Clean | FGSM | PGD | C&W | AA |
|---|---|---|---|---|---|
| ARD | 85.57 | 61.07 | 52.13 | 50.72 | 48.78 |
| IAD | 84.64 | 60.78 | 53.10 | 50.73 | 48.67 |
| RSLAD | 84.12 | 60.37 | 54.68 | 51.96 | 50.58 |
| AKD | 84.51 | 60.12 | 52.17 | 50.71 | 49.03 |
| AdaAD | 85.10 | 61.89 | 56.57 | 53.59 | 52.43 |
| IGDM | 85.31 | 62.90 | 57.28 | 53.91 | 52.55 |
| SAAD | 85.78 | 62.77 | 57.25 | 53.83 | 52.69 |

## 5.3 Generalization and Ablation Studies

**Adversarial Distillation with ERT.** Table 7 demonstrates SAAD's performance in a high-transferability scenario using an ERT. In this setting, SAAD matches or slightly exceeds IGDM. This finding highlights a key aspect of our method: even though SAAD's primary mechanism targets low-transferability, it maintains strong performance in this favorable scenario—a crucial feature given that a teacher's effectiveness is often unknown in practice. Therefore, this result validates SAAD as a robust default: it incurs no performance degradation in favorable settings (with an ERT) while significantly improving robustness when transferability is weak (with an IRT).

**OODRobustBench Evaluation.** To evaluate the generalization of adversarial robustness beyond in-distribution test sets, we additionally conduct experiments on OODRobustBench (Li et al., 2024), a benchmark specifically designed to assess robustness under distribution shifts. It includes two major types of shifts: dataset shifts and threat shifts. For $OOD_d$, we report both the clean accuracy on naturally shifted datasets (C-$OOD_d$) and the robust accuracy under MM5 adversarial attacks (Gao et al., 2022) (R-$OOD_d$), which encompass corruptions such as noise and blur. On the other hand, $OOD_t$ evaluates robustness against six unseen attack types, including both large-$\epsilon$ $l_p$-norm attacks and non-$l_p$ threat models. As shown in Table 8, SAAD consistently outperforms existing AD methods across both dataset and threat shifts.

**Black-box Attack Evaluation.** We further evaluate the students against query-based black-box attacks to verify their robustness in scenarios where gradient information is unavailable. We employ RayS (Chen & Gu, 2020), Square Attack (Andriushchenko et al., 2020), and SPSA (Uesato et al., 2018) using the

Table 8: OODRobustBench results and Black-box attack robustness on CIFAR-10. All methods are distilled from the `Gowal2021Improving` teacher to ResNet-18 student.

| Method | AA | OODRobustBench | | | Black-box Attacks | | |
|---|---|---|---|---|---|---|---|
| | | C-OOD$_d$ | R-OOD$_d$ | OOD$_t$ | RayS(40k) | Square(5k) | SPSA(40k) |
| ARD | 40.79 | 73.92 | 25.87 | 18.25 | 46.42 | 49.57 | 51.26 |
| IAD | 40.70 | 73.80 | 25.97 | 18.63 | 46.49 | 49.50 | 51.28 |
| RSLAD | 40.57 | 74.97 | 27.25 | 20.77 | 48.21 | 50.89 | 52.93 |
| AKD | 40.95 | 73.87 | 25.99 | 18.25 | 46.82 | 49.68 | 50.95 |
| AdaAD | 43.27 | 74.36 | 27.43 | 20.62 | 48.68 | 52.08 | 53.44 |
| IGDM | 44.76 | 71.19 | 30.40 | 24.83 | 49.67 | 52.01 | 53.83 |
| **SAAD-C** | 49.72 | **76.34** | 34.52 | 26.06 | **55.57** | **58.96** | **60.19** |
| **SAAD** | **50.35** | 74.47 | **35.36** | **27.19** | 55.47 | 58.61 | 60.00 |

ResNet-18 students distilled from the `Gowal2021Improving` teacher. As presented in Table 8, SAAD exhibits substantially stronger robustness than other distillation methods in this setting. The fact that SAAD maintains high accuracy against these diverse query-based attacks confirms that our method effectively defends against threats regardless of the attacker's access to model gradients.

# 6 Conclusion

In this paper, we challenged the common assumption that a more robust teacher necessarily yields a more robust student in adversarial distillation. We showed that the key bottleneck is not the capacity gap but the transferability of adversarial perturbations between teacher and student. To address this, we introduced Sample-wise Adaptive Adversarial Distillation (SAAD), which dynamically up-weights those examples whose adversarial attacks on the student remain effective for the teacher, and proposed a complementary clean distillation variant (SAAD-C) that provides a tunable, favorable trade-off to recover clean accuracy from non-transferable samples. We experimentally demonstrated that our approach consistently outperforms existing AD methods and even standard adversarial training when transferability is limited.

# Acknowledgement

This work was supported by the National Research Foundation of Korea (NRF) grant funded by the Korea government (MSIT) (No. RS-2024-00408003 and RS-2025-00516153) and the Institute for Information & communications Technology Planning & Evaluation (IITP) grant funded by the Korea government (MSIT) (No. RS-2024-00444862 and RS-2026-25522672).

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

# A  Implementation Details

## A.1  Robust Teacher and Analysis Details

Table 9 summarizes the teacher models used in our study, all selected from RobustBench (Croce et al., 2021). Table 10 presents the results of adversarial distillation from each teacher into a ResNet-18 student, using six recent distillation methods. For each teacher, we report the student's AutoAttack accuracy under each method, the average accuracy across methods (Mean AA), and the transferable adversarial sample ratio (TAS). As introduced in Section 3, we divide teachers into two groups for clear interpretation: *Effective Robust Teachers* (ERTs) and *Ineffective Robust Teachers* (IRTs). This categorization is based on the Mean AA value: a teacher is labeled as an ERT if the Mean AA exceeds the TRADES baseline (46.46%) by more than 3 percentage points, and as an IRT if it falls below that baseline by more than 3 points. The $\pm 3\%$ band and the ERT/IRT split are used only in Section 3 for analysis, to highlight clearly separated cases. All analyses presented in the main paper—including TAS, adversarial variance (AVar), robust overfitting (RO), and baseline comparisons in Table 5—are conducted using RSLAD as the baseline AD method.

Table 9: Summary of teacher models used in our study, selected from RobustBench. 'Abbr.' denotes the shorthand identifier used in Figure 1. **Bolded entries** correspond to the teachers analyzed in Section 3; in that section, their RobustBench names are shown without architecture suffixes (e.g., `Bartoldson2024Adversarial_WRN-94-16` is referred to as `Bartoldson2024Adversarial`)

| Abbr. | RobustBench name | Architecture | Size(M) | Clean | AA |
|---|---|---|---|---|---|
| Bart94 | **Bartoldson2024Adversarial_WRN-94-16** (Bartoldson et al., 2024) | WRN-94-16 | 365.92 | 93.68 | 73.71 |
| Bart82 | Bartoldson2024Adversarial_WRN-82-8 (Bartoldson et al., 2024) | WRN-82-8 | 79.13 | 93.11 | 71.59 |
| Wang70 | Wang2023Better_WRN-70-16 (Wang et al., 2023c) | WRN-70-16 | 266.80 | 93.25 | 70.69 |
| Cui28 | Cui2023Decoupled_WRN-28-10 (Cui et al., 2023) | WRN-28-10 | 36.48 | 92.16 | 67.73 |
| Gowal70 | Gowal2020Uncovering_70_16_extra (Gowal et al., 2020) | WRN-70-16 | 266.80 | 91.10 | 65.87 |
| Rebu70 | **Rebuffi2021Fixing_70_16_cutmix_ddpm** (Rebuffi et al., 2021) | WRN-70-16 | 266.80 | 88.54 | 64.20 |
| Gowal28 | **Gowal2021Improving_28_10_ddpm_100m** (Gowal et al., 2021) | WRN-28-10 | 36.48 | 87.50 | 63.38 |
| Huang | Huang2021Exploring_ema (Huang et al., 2021) | WRN-34-R | 68.12 | 91.23 | 62.54 |
| Dai | Dai2021Parameterizing (Dai et al., 2022) | WRN-28-10 | 36.48 | 87.02 | 61.55 |
| Sridhar34 | Sridhar2021Robust_34_15 (Sridhar et al., 2022) | WRN-34-15 | 108.53 | 86.53 | 60.41 |
| Carmon | Carmon2019Unlabeled (Carmon et al., 2019) | WRN-28-10 | 36.48 | 89.69 | 59.53 |
| GowalR18 | Gowal2021Improving_R18_ddpm_100m (Gowal et al., 2021) | PreActRN-18 | 12.55 | 87.35 | 58.50 |
| Chen34-20 | Chen2021LTD_WRN34_20 (Chen & Lee, 2021) | WRN-34-20 | 184.53 | 86.03 | 57.71 |
| Chen34-10 | **Chen2021LTD_WRN34_10** (Chen & Lee, 2021) | WRN-34-10 | 46.16 | 85.21 | 56.94 |
| SehwagR18 | Sehwag2021Proxy_R18 (Sehwag et al., 2021) | RN-18 | 11.17 | 84.59 | 55.54 |

## A.2  Adversarial Variance Details

In this section, we provide further technical details on the computation of adversarial variance and the associated decomposition of adversarial error. Algorithm 1 outlines the procedure used throughout the main paper to estimate adversarial variance under AD. This algorithm quantifies how much the learned student model varies when trained on different subsets of the training data, using a fixed AD method. As shown in the algorithm, we split the full training dataset $\mathcal{D}$ into $N$ disjoint subsets and independently train student models on each subset using a fixed AD method. This allows us to observe how much the resulting models vary in their outputs under adversarial evaluation. For each trained student, the variation is measured by evaluating each model's prediction at a fixed test point $(\mathbf{x}, \mathbf{y})$ under its corresponding adversarial input, and computing the KL divergence between these predictions and their geometric mean, reflecting how much model outputs fluctuate due to data-induced randomness. Following (Yu et al., 2021), we set the number of splits $N = 2$, corresponding to training each student on half of CIFAR-10 (25,000 examples). To obtain more stable estimates, we repeat this procedure $K = 2$ times with different random splits.

To further clarify the theoretical interpretation of our measure, we restate and prove a bias–variance decomposition of the expected adversarial error, following formulations introduced in prior works (Yu et al., 2021; Zhou et al., 2021). We explicitly show how the adversarial prediction error decomposes into three terms: intrinsic noise, adversarial bias, and adversarial variance. This derivation justifies our use of KL-

Table 10: AutoAttack accuracy (%) of student models distilled from each RobustBench teacher using different methods. Each row corresponds to a teacher model shown, and columns represent distillation methods in Figure 1. Mean AA denotes the average performance across all methods for a given teacher. TAS indicates the transferable adversarial sample ratio with RSLAD method. **Bold Mean AA** values indicate ERTs, whose students outperform the TRADES baseline (46.46%) by more than 3 percentage points. Underlined Mean AA values denote IRTs, whose students fall short of the baseline by more than 3 points.

| Abbr. | ARD | IAD | RSLAD | AKD | AdaAD | IGDM | Mean AA | TAS |
|---|---|---|---|---|---|---|---|---|
| Bart94 | 41.94 | 41.83 | 44.07 | 41.73 | 44.68 | 44.75 | 43.17 | 0.1991 |
| Bart82 | 44.93 | 45.07 | 47.59 | 44.02 | 48.63 | 48.80 | 46.51 | 0.3779 |
| Wang70 | 44.59 | 44.75 | 47.38 | 44.53 | 48.47 | 48.66 | 46.40 | 0.3656 |
| Cui28 | 46.16 | 46.20 | 49.07 | 45.87 | 50.50 | 50.79 | 48.10 | 0.5400 |
| Gowal70 | 45.88 | 46.84 | 48.89 | 45.82 | 50.82 | 50.82 | 48.18 | 0.5774 |
| Rebu70 | 47.75 | 47.95 | 50.94 | 48.11 | 52.14 | 52.28 | **50.20** | 0.6770 |
| Gowal28 | 40.94 | 40.85 | 41.08 | 41.13 | 43.01 | 44.26 | 42.05 | 0.1494 |
| Huang | 46.00 | 45.81 | 49.09 | 45.26 | 49.20 | 49.26 | 47.44 | 0.4431 |
| Dai | 40.64 | 40.52 | 42.28 | 40.61 | 42.87 | 43.92 | 41.81 | 0.2175 |
| Sridhar34 | 47.72 | 46.89 | 49.89 | 46.29 | 51.50 | 52.26 | 48.84 | 0.8133 |
| Carmon | 46.16 | 45.94 | 49.56 | 46.56 | 51.26 | 51.53 | 48.50 | 0.6450 |
| GowalR18 | 44.12 | 43.62 | 46.61 | 44.41 | 49.51 | 50.46 | 46.72 | 0.4213 |
| Chen34-20 | 48.78 | 48.67 | 50.58 | 49.03 | 52.43 | 52.55 | **50.34** | 0.9262 |
| Chen34-10 | 50.82 | 50.55 | 52.21 | 50.95 | 53.24 | 53.45 | **51.87** | 0.9810 |
| SehwagR18 | 42.30 | 42.28 | 45.33 | 43.65 | 48.95 | 49.69 | 45.37 | 0.3689 |

based adversarial variance as a meaningful quantity for analyzing robustness under adversarial training and distillation.

We aim to show that the expected adversarial error satisfies the following lemma:

**Lemma 1** (Decomposition of Expected Adversarial Error). *Let* $(\mathbf{x}, \mathbf{y})$ *be a test example, where* $\mathbf{y} = [y_1, \ldots, y_C]^\top \in \Delta^{C-1}$ *is the target class–probability vector over* $C$ *classes. Let* $\mathcal{D}$ *denote the training dataset. Define the adversarial prediction*

$$\hat{\mathbf{y}} \;=\; f_{\hat{\boldsymbol{\theta}}(\mathcal{D})}\big(\mathbf{x} + \boldsymbol{\delta}(\mathbf{x}, \mathbf{y}, \mathcal{D})\big), \tag{10}$$

*where* $\boldsymbol{\delta}(\mathbf{x}, \mathbf{y}, \mathcal{D})$ *is the worst–case perturbation within the chosen threat model. Let the* normalized geometric mean *of predictions across* $\mathcal{D}$ *be*

$$\bar{\mathbf{y}} \;:=\; \frac{1}{Z(\mathbf{x})} \exp\big(\mathbb{E}_{\mathcal{D}}[\log \hat{\mathbf{y}}]\big), \qquad Z(\mathbf{x}) \text{ chosen so } \sum_{c=1}^{C} \bar{y}_c = 1. \tag{11}$$

*The expected adversarial cross–entropy*

$$\mathrm{CE}(\mathbf{y}, \hat{\mathbf{y}}) \;:=\; -\sum_{c=1}^{C} y_c \log \hat{y}_c \tag{12}$$

*admits the decomposition*

$$\mathbb{E}_{\mathbf{x}, \mathcal{D}}\left[\mathrm{CE}(\mathbf{y}, \hat{\mathbf{y}})\right] = \underbrace{\mathbb{E}_{\mathbf{x}}\left[-\mathbf{y}\log\mathbf{y}\right]}_{\textit{Intrinsic Noise}} + \underbrace{\mathbb{E}_{\mathbf{x}}\left[\mathbf{y}\log\frac{\mathbf{y}}{\bar{\mathbf{y}}}\right]}_{\textit{Adversarial Bias}} + \underbrace{\mathbb{E}_{\mathbf{x}, \mathcal{D}}\left[\mathrm{KL}(\bar{\mathbf{y}}\,\|\,\hat{\mathbf{y}})\right]}_{\textit{Adversarial Variance}}. \tag{13}$$

*Proof.* Fix a test point $(\mathbf{x}, \mathbf{y})$. By definition,

$$\mathrm{CE}(\mathbf{y}, \hat{\mathbf{y}}) = -\mathbf{y}\log\hat{\mathbf{y}}. \tag{14}$$

---

**Algorithm 1** Estimating Adversarial Variance under Adversarial Distillation

---

**Require:** Test point $(\mathbf{x}, \mathbf{y})$, dataset $\mathcal{D} = \{(\mathbf{x}_i, \mathbf{y}_i)\}_{i=1}^n$ , teacher $f_T$, number of splits $N$, repetitions $K$

1: **for** $k = 1$ **to** $K$ **do**

2:     Randomly split $\mathcal{D}$ into $\{\mathcal{D}_j^{(k)}\}_{j=1}^N$.

3:     **for** $j = 1$ **to** $N$ **do**

4:         Adversarially distill student $f_S(\,\cdot\,; \boldsymbol{\theta})$ on $\mathcal{D}_j^{(k)}$ with $f_T$:

$$\hat{\boldsymbol{\theta}}\big(\mathcal{D}_j^{(k)}\big) \;\approx\; \arg\min_{\boldsymbol{\theta}} \; \frac{1}{|\mathcal{D}_j^{(k)}|} \sum_{i \in \mathcal{D}_j^{(k)}} \Big[ \max_{\boldsymbol{\delta}_{S,i} \in \Delta} L_{\mathrm{AD}}(f_S, f_T, \mathbf{x}_i, \boldsymbol{\delta}_{S,i}) \Big].$$

5:         Find adversarial perturbation $\boldsymbol{\delta}_j := \boldsymbol{\delta}(\mathbf{x}, \mathbf{y}, \mathcal{D}_j^{(k)})$ that approximately solves

$$\max_{\boldsymbol{\delta} \in \Delta} L_{\max}\big(f_{\hat{\boldsymbol{\theta}}(\mathcal{D}_j^{(k)})}(\mathbf{x} + \boldsymbol{\delta})\,,\, \mathbf{y}\big),$$

6:         Evaluate the adversarial student prediction $\hat{\mathbf{y}}_j := f_S(\mathbf{x} + \boldsymbol{\delta}_j; \hat{\boldsymbol{\theta}}(\mathcal{D}_j^{(k)}))$.

7:     **end for**

8:     Aggregate via the geometric mean on the simplex:

$$\bar{\mathbf{y}} := \frac{1}{Z} \exp\left( \frac{1}{N} \sum_{j=1}^N \log \hat{\mathbf{y}}_j \right), \quad Z \text{ is a normalization constant.}$$

9:     Compute the KL-based variance for split $k$:

$$\widehat{\mathrm{AVar}}_{\mathrm{KL}}\big(\mathbf{x}, \mathbf{y}, \mathcal{D}^{(k)}\big) = \frac{1}{N} \sum_{j=1}^N \mathrm{KL}\big(\bar{\mathbf{y}} \,\|\, \hat{\mathbf{y}}_j\big).$$

10: **end for**

11: **Return** the averaged adversarial variance

$$\widehat{\mathrm{AVar}}_{\mathrm{KL}}(\mathbf{x}, \mathbf{y}) = \frac{1}{K} \sum_{k=1}^K \widehat{\mathrm{AVar}}_{\mathrm{KL}}\big(\mathbf{x}, \mathbf{y}, \mathcal{D}^{(k)}\big).$$

---

We add and subtract the term $\mathbf{y} \log \bar{\mathbf{y}}$, yielding:

$$\mathrm{CE}(\mathbf{y}, \hat{\mathbf{y}}) = -\mathbf{y} \log \hat{\mathbf{y}} = -\mathbf{y} \log \mathbf{y} + \mathbf{y} \log \frac{\mathbf{y}}{\bar{\mathbf{y}}} + \mathbf{y} \log \frac{\bar{\mathbf{y}}}{\hat{\mathbf{y}}}. \tag{15}$$

Taking expectation over $\mathcal{D}$ on both sides gives:

$$\mathbb{E}_{\mathcal{D}}\big[\mathrm{CE}(\mathbf{y}, \hat{\mathbf{y}})\big] = -\mathbf{y} \log \mathbf{y} + \mathbf{y} \log \frac{\mathbf{y}}{\bar{\mathbf{y}}} + \mathbb{E}_{\mathcal{D}}\Big[\mathbf{y} \log \frac{\bar{\mathbf{y}}}{\hat{\mathbf{y}}}\Big]. \tag{16}$$

The first term corresponds to the intrinsic noise, the second to adversarial bias, and it remains to show that the third term equals the adversarial variance:

$$\mathbb{E}_{\mathcal{D}}\Big[\mathbf{y} \log \frac{\bar{\mathbf{y}}}{\hat{\mathbf{y}}}\Big] = \mathbb{E}_{\mathcal{D}}\big[\mathrm{KL}(\bar{\mathbf{y}} \| \hat{\mathbf{y}})\big]. \tag{17}$$

To see this, recall that $\log \bar{\mathbf{y}} = \mathbb{E}_{\mathcal{D}}[\log \hat{\mathbf{y}}] - \log Z$ component-wise. For each class $c$,

$$\mathbb{E}_{\mathcal{D}}\Big[y_c \log \frac{\bar{y}_c}{\hat{y}_c}\Big] = y_c \left(\mathbb{E}_{\mathcal{D}}[\log \hat{y}_c] - \log Z\right) - y_c \, \mathbb{E}_{\mathcal{D}}[\log \hat{y}_c] = -y_c \log Z. \tag{18}$$

Summing over all classes gives:

$$\mathbb{E}_{\mathcal{D}}\Big[\mathbf{y}\log\frac{\bar{\mathbf{y}}}{\hat{\mathbf{y}}}\Big] = -\log Z\sum_c y_c = -\log Z. \tag{19}$$

On the other hand:

$$\begin{aligned}
\mathbb{E}_{\mathcal{D}}[\mathrm{KL}(\bar{\mathbf{y}}\|\hat{\mathbf{y}})] &= \mathbb{E}_{\mathcal{D}}\left[\sum_c \bar{y}_c\log\frac{\bar{y}_c}{\hat{y}_c}\right] \\
&= \sum_c \bar{y}_c\left(\mathbb{E}_{\mathcal{D}}[\log\hat{y}_c] - \log Z - \mathbb{E}_{\mathcal{D}}[\log\hat{y}_c]\right) \\
&= -\log Z\sum_c \bar{y}_c = -\log Z.
\end{aligned} \tag{20}$$

Since $\sum_c y_c = \sum_c \bar{y}_c = 1$, the two expressions are equal, completing the proof. $\qquad\square$

### A.3  Proposed Method Details

### A.3.1  From TAS to the surrogate loss

Let $C$ be the number of classes and $\Delta^{C-1}$ the probability simplex. For an input $\mathbf{x}$, define the teacher's predictive distributions under student- and teacher-crafted adversarial perturbations by

$$\boldsymbol{p}(\mathbf{x}) \coloneqq f_T(\mathbf{x}+\boldsymbol{\delta}_S)\in\Delta^{C-1}, \qquad \boldsymbol{q}(\mathbf{x}) \coloneqq f_T(\mathbf{x}+\boldsymbol{\delta}_T)\in\Delta^{C-1}, \tag{21}$$

with components $p_i = [\boldsymbol{p}(\mathbf{x})]_i$ and $q_i = [\boldsymbol{q}(\mathbf{x})]_i$. We define the TAS score by

$$\mathrm{TAS}(\mathbf{x}) \coloneqq \mathrm{KL}\big(\boldsymbol{p}(\mathbf{x})\,\|\,f_T(\mathbf{x})\big) - \mathrm{KL}\big(\boldsymbol{p}(\mathbf{x})\,\|\,\boldsymbol{q}(\mathbf{x})\big). \tag{22}$$

We call $\mathbf{x}$ a transferable adversarial sample if $\mathrm{TAS}(\mathbf{x})\geq 0$, which is equivalent to

$$\mathrm{KL}\big(f_T(\mathbf{x}+\boldsymbol{\delta}_S)\,\|\,f_T(\mathbf{x})\big) \geq \mathrm{KL}\big(f_T(\mathbf{x}+\boldsymbol{\delta}_S)\,\|\,f_T(\mathbf{x}+\boldsymbol{\delta}_T)\big). \tag{23}$$

In the main text, for analysis, we use $\mathrm{TAS}(\mathbf{x})\geq 0$ to decide whether a sample is TAS; otherwise it is non-TAS. It cleanly separates samples and reveals how the two groups differ in (i) the entropy of teacher logits on student-crafted adversarial inputs (Figure 4a), (ii) adversarial variance when training on each subset after a warm-up phase (Figure 4b), and (iii) robust overfitting trajectories (Figure 4c). These results show that the non-TAS subset concentrates low-entropy, high-variance supervision and drives robust overfitting, whereas the TAS subset exhibits higher entropy and more stable generalization.

While the binary split is useful for diagnostics, transferability is not inherently binary for training: samples lie at different distances from the boundary $\mathrm{TAS}(\mathbf{x})=0$, stochasticity near that boundary can flip membership, and discarding non-TAS samples is data-inefficient. Therefore, for training we replace the binary split by a continuous score, which we map to sample weights as in equation 8.

For training-time efficiency, we avoid computing the teacher-side perturbation $\boldsymbol{\delta}_T$ and instead use an entropy-based proxy on $f_T(\mathbf{x}+\boldsymbol{\delta}_S)$ in equation 8. Assuming a strong white-box $\boldsymbol{\delta}_T$ yields a high-entropy, non-degenerate teacher distribution $\boldsymbol{q}(\mathbf{x}) = f_T(\mathbf{x}+\boldsymbol{\delta}_T)$ (so $m = \min_i q_i > 0$), we have:

**Lemma 2** (Entropy-based lower bound on TAS)**.** *Assume the teacher's adversarial output satisfies $m = \min_i q_i > 0$, where $\boldsymbol{q}(\mathbf{x}) = f_T(\mathbf{x}+\boldsymbol{\delta}_T)$ and $q_i = [\boldsymbol{q}(\mathbf{x})]_i$. Then*

$$\mathrm{TAS}(\mathbf{x}) \geq H(f_T(\mathbf{x}+\boldsymbol{\delta}_S)) + \log m. \tag{24}$$

*Proof.* Let $H(\boldsymbol{p}) = -\sum_i p_i\log p_i$. By the definition of KL divergence,

$$\mathrm{KL}(\boldsymbol{p}\|\boldsymbol{q}) = -H(\boldsymbol{p}) - \sum_i p_i\log q_i \leq -H(\boldsymbol{p}) - \log m. \tag{25}$$

Table 11: Comparison of inner maximization ($L_{\max}$), outer minimization ($L_{\min}$) for various AD methods. As IGDM follows a modular design, $L_{\mathrm{AD}}$ can be any other AD outer minimization loss.

| Method | Inner Maximization | Outer Minimization |
|---|---|---|
| ARD | $\mathrm{CE}\big(\mathbf{y}, f_S(\mathbf{x}+\boldsymbol{\delta})\big)$ | $\mathrm{KL}\big(f_T(\mathbf{x})\|f_S(\mathbf{x}+\boldsymbol{\delta})\big)$ |
| RSLAD | $\mathrm{KL}\big(f_T(\mathbf{x})\|f_S(\mathbf{x}+\boldsymbol{\delta})\big)$ | $\mathrm{KL}\big(f_T(\mathbf{x})\|f_S(\mathbf{x}+\boldsymbol{\delta})\big)$ |
| AdaAD | $\mathrm{KL}\big(f_T(\mathbf{x}+\boldsymbol{\delta})\|f_S(\mathbf{x}+\boldsymbol{\delta})\big)$ | $\mathrm{KL}\big(f_T(\mathbf{x}+\boldsymbol{\delta})\|f_S(\mathbf{x}+\boldsymbol{\delta})\big)$ |
| IGDM | $\mathrm{KL}\big(f_T(\mathbf{x}+\boldsymbol{\delta})\|f_S(\mathbf{x}+\boldsymbol{\delta})\big)$ | $L_{\mathrm{AD}}+\alpha_{\mathrm{IGDM}}\cdot\mathrm{KL}\big(f_T(\mathbf{x}+\boldsymbol{\delta})-f_T(\mathbf{x}-\boldsymbol{\delta})\|f_S(\mathbf{x}+\boldsymbol{\delta})-f_S(\mathbf{x}-\boldsymbol{\delta})\big)$ |

Since $\mathrm{TAS}(\mathbf{x}) = \mathrm{KL}\big(\boldsymbol{p}\|f_T(\mathbf{x})\big) - \mathrm{KL}\big(\boldsymbol{p}\|\boldsymbol{q}\big)$, we obtain

$$\mathrm{TAS}(\mathbf{x}) \ \geq\ H(\boldsymbol{p}) + \log m = H(f_T(\mathbf{x}+\boldsymbol{\delta}_S)) + \log m. \tag{26}$$

$\square$

While this bound provides conceptual motivation, we acknowledge that for overconfident teachers, $m$ can be extremely small, making the lower bound vacuous in practice. Therefore, rather than a strict formal justification, we adopt this entropy-based formulation as an empirically motivated heuristic for sample-level transferability: higher teacher entropy empirically correlates with higher transferability, providing a computation-friendly proxy without evaluating $\boldsymbol{\delta}_T$.

### A.3.2 Revisiting Previous Adversarial Distillation Methods

Existing AD methods have largely been designed and evaluated under ERTs, with limited analysis conducted in the context of IRTs. While overall distillation performance is not effective under IRTs, as shown in Figure 1, Table 4, and Table 6, our analysis reveals that some AD methods still perform comparatively better than other AD methods even under IRTs. Table 11 summarizes representative AD methods, focusing on their inner maximization and outer minimization formulations. The differences in these formulations determine how each method responds to the adversarial signal provided by the teacher. Among them, IGDM further refines robustness alignment by implicitly matching gradients (adversarial direction) through logit difference minimization in the outer optimization. This design can be interpreted as encouraging transferability between the student and teacher. Empirically, IGDM consistently yields improved robustness compared to prior methods, particularly under IRTs, which motivates our decision to adopt IGDM as our baseline. Since IGDM is a modular design rather than a complete distillation framework, we adopt it in combination with AdaAD, following the original implementation (Lee et al., 2025). Therefore, the overall SAAD minimization loss is defined as:

$$L_{\mathrm{SAAD}} = \frac{1}{N} \sum_{i=1}^{N} w_i \cdot L_{\mathrm{AD}}(f_S, f_T, \mathbf{x}_i, \boldsymbol{\delta}_i), \quad w_i := H\left(f_T(\mathbf{x}_i + \boldsymbol{\delta}_i)\right), \tag{27}$$

where $H(\cdot)$ denotes the entropy function, and $L_{\mathrm{AD}}$ represents the base adversarial distillation loss using AdaAD with IGDM, formulated as:

$$\begin{aligned} L_{\mathrm{AD}}(f_S, f_T, \mathbf{x}_i, \boldsymbol{\delta}_i) = \ &\mathrm{KL}\left(f_T(\mathbf{x}_i + \boldsymbol{\delta}_i) \,\|\, f_S(\mathbf{x}_i + \boldsymbol{\delta}_i)\right) \\ &+ \alpha_{\mathrm{IGDM}} \cdot \mathrm{KL}\left(f_T(\mathbf{x}_i + \boldsymbol{\delta}_i) - f_T(\mathbf{x}_i) \,\|\, f_S(\mathbf{x}_i + \boldsymbol{\delta}_i) - f_S(\mathbf{x}_i)\right). \end{aligned} \tag{28}$$

In results, our baseline implementation follows AdaAD with the IGDM module, along with an weight averaging (SWA) (Izmailov et al., 2018).

### A.3.3 Fast Inner Maximization via Linear Approximation

We also introduce a lightweight inner maximization strategy to reduce computational cost. This technique is a practical enhancement to the SAAD framework by avoiding repeated teacher backpropagation, enabling

---

**Algorithm 2** Fast Inner Maximization with First-Order Teacher Logit Correction

---

**Require:** Input $\mathbf{x}$, true label $y$, student $f_S$, teacher $f_T$, step size $\eta$, perturbation bound $\epsilon$, steps $K$, correction weight $\lambda_{\text{in}}$

1: Compute teacher logits on $\mathbf{x}$: $\ell_T$
2: Compute gradient $\nabla_{\mathbf{x}}\ell_T^y(\mathbf{x})$, the input gradient of the logit corresponding to class $y$
3: Initialize perturbation: $\boldsymbol{\delta} \leftarrow 0.001 \cdot \mathcal{N}(0, I)$
4: **for** $k = 1$ to $K$ **do**
5:  Compute correction term:
$$\Delta\ell^y \leftarrow \lambda_{\text{in}} \cdot \langle \nabla_{\mathbf{x}}\ell_T^y(\mathbf{x}), \boldsymbol{\delta} \rangle$$
6:  Construct corrected logits: $\tilde{\ell}_T \leftarrow \ell_T$, with $\tilde{\ell}_T^y \leftarrow \ell_T^y + \Delta\ell^y$
7:  Compute loss: $\mathcal{L}_{\text{KL}} := \text{KL}\left(\text{softmax}(\tilde{\ell}_T) \,\|\, f_S(\mathbf{x} + \boldsymbol{\delta})\right)$
8:  Update perturbation:
$$\boldsymbol{\delta} \leftarrow \text{Proj}_{\boldsymbol{\delta} \in \Delta}\left(\boldsymbol{\delta} + \eta \cdot \text{sign}\left(\nabla_{\boldsymbol{\delta}}\mathcal{L}_{\text{KL}}\right)\right)$$
9:  Project $\mathbf{x} + \boldsymbol{\delta} \in [0, 1]^d$
10: **end for**
11: **return** $\mathbf{x} + \boldsymbol{\delta}$

---

efficient training without compromising robustness. Recent AD methods have adopted iterative inner maximization procedures involving teacher backpropagation (Huang et al., 2023; Lee et al., 2025), a strategy that has empirically led to stronger robustness in distilled students. However, as the size of the teacher model increases, this process becomes computationally expensive. To alleviate this cost, we introduce an approximation technique that reduces the number of teacher backpropagations from multiple iterations to a single step. Inspired by the locally linear behavior of adversarially trained models discussed in prior work (Lee et al., 2025), we linearize the teacher's output around the input $\mathbf{x}$ using a first-order Taylor expansion:

$$f_T(\mathbf{x} + \boldsymbol{\delta}) \approx f_T(\mathbf{x}) + \langle \nabla_{\mathbf{x}} f_T(\mathbf{x}), \boldsymbol{\delta} \rangle, \tag{29}$$

and substitute this into the inner maximization objective:

$$\max_{\boldsymbol{\delta} \in \Delta} \text{KL}\left(f_T(\mathbf{x} + \boldsymbol{\delta}) \,\|\, f_S(\mathbf{x} + \boldsymbol{\delta})\right), \tag{30}$$

which results in the following approximation:

$$\max_{\boldsymbol{\delta} \in \Delta} \text{KL}\left(f_T(\mathbf{x} + \boldsymbol{\delta}) \,\|\, f_S(\mathbf{x}) + \langle \nabla_{\mathbf{x}} f_S(\mathbf{x}), \boldsymbol{\delta} \rangle\right). \tag{31}$$

This formulation enables a technically efficient alternative to conventional inner maximization by bypassing the need for iterative teacher backpropagation while preserving the gradient-guided adversarial direction. Algorithm 2 details the procedure. Given a clean input and a true label, we first compute the teacher logits and the corresponding input gradient. At each step, the KL divergence is computed between the student output and the corrected teacher logits. By adopting this approximation, we achieve a nearly $4\times$ speed-up in the inner maximization step compared to the original AdaAD formulation in Table 17, while maintaining comparable robustness and distillation performance.

### A.3.4 Main Algorithm

The overall training procedure for SAAD and SAAD-C is outlined in Algorithm 3.

## B Experimental Details

### B.1 Adversarial Distillation Settings

We conduct experiments on CIFAR-10, CIFAR-100, and Tiny-ImageNet, applying standard data augmentations (random cropping and horizontal flipping). All student models are trained for 200 epochs using SGD

---

**Algorithm 3** Training Algorithm for SAAD and SAAD-C

---

**Require:** Teacher $f_T$, student $f_S$, training set $\mathcal{D}$, step size $\eta$, total epochs $E$, perturbation bound $\epsilon$, inner steps $K$, weights $\lambda_{\text{in}}, \alpha_{\text{IGDM}}, \beta$

1: **for** epoch = 1 to $E$ **do**
2:     **for** each minibatch $\{(\mathbf{x}_i, y_i)\}_{i=1}^{B} \sim \mathcal{D}$ **do**
3:         Generate $\boldsymbol{\delta}_i$ for each $\mathbf{x}_i$ using Algorithm 2
4:         Compute per-sample entropy: $w_i \leftarrow H(f_T(\mathbf{x}_i + \boldsymbol{\delta}_i))$
5:         Normalize: $\hat{w}_i \leftarrow \text{Normalize}(w_i)$
6:         Compute loss $L_{\text{AD}}$ as in equation 28
7:         Compute total loss for SAAD equation 8:

$$\ell_i \leftarrow w_i \cdot L_{\text{AD}}(\mathbf{x}_i, \boldsymbol{\delta}_i)$$

8:         **if** SAAD-C **then**
9:             Add clean distillation:

$$\ell_i \leftarrow \ell_i + \beta \cdot (1 - \hat{w}_i) \cdot \text{KL}(f_T(\mathbf{x}_i) \,\|\, f_S(\mathbf{x}_i))$$

10:         **end if**
11:         $\mathcal{L} \leftarrow \frac{1}{B} \sum_{i=1}^{B} \ell_i$
12:         Update $f_S$ via gradient descent: $\boldsymbol{\theta} \leftarrow \boldsymbol{\theta} - \eta \nabla_{\boldsymbol{\theta}} \mathcal{L}$
13:     **end for**
14:     **if** epoch $\geq 95$ **then**
15:         Update SWA parameters
16:     **end if**
17: **end for**

---

with momentum 0.9, weight decay $5 \times 10^{-4}$, and an initial learning rate of 0.1, which is decayed by a factor of 10 at the 100th and 150th epochs. The batch size is fixed to 128 across all experiments.

We compare two adversarial training baselines—PGD-AT and TRADES—alongside six recent adversarial distillation methods: ARD, IAD, RSLAD, AKD, AdaAD, and IGDM. As IGDM is a modular technique, we evaluate its performance in conjunction with AdaAD (i.e., AdaAD+IGDM), which consistently yields the best results among IGDM-integrated variants. Following the original design, we set the IGDM hyperparameter $\alpha_{\text{IGDM}}$ to 1 for CIFAR-10, 20 for CIFAR-100, and 10 for Tiny-ImageNet. This configuration is also adopted in our SAAD framework to ensure consistency across evaluations.

For inner maximization in training, we adopt a standard multi-step attack setup with $L_\infty$ perturbations bounded by $\epsilon = 8/255$, step size $2/255$, and 10 iterations. Each method follows its original inner maximization formulation, summarized in Table 11. Specifically, PGD-AT, ARD, IAD, and AKD use student-only cross-entropy loss; TRADES employs KL divergence between clean and perturbed predictions; RSLAD aligns student outputs with teacher predictions on clean inputs; while AdaAD and IGDM match student and teacher predictions under shared perturbations. Our proposed SAAD framework follows the inner-outer structure described in Algorithm 2.

## B.2   Selected Hyperparameters

We document the selected values of the clean distillation weighting coefficient $\beta$, which controls the strength of the clean KL divergence term in SAAD-C. To determine $\beta$, we perform a small-scale grid search aiming to maximize clean accuracy while maintaining comparable AutoAttack robustness to the SAAD. On CIFAR-10, we set $\beta = 0.2$ across all combinations of ResNet-18 and MobileNetV2 students with either `Bartoldson2024Adversarial` or `Gowal2021Improving` teachers. On CIFAR-100 and Tiny-ImageNet, we apply a uniform setting of $\beta = 0.5$.

### B.3 Additional Notes on Figures and Tables

**Figure 1a.** All teacher models consistently achieve AA accuracies above 55%, while the strongest student reaches only 54%. This ensures that student underperformance cannot be attributed to weak teacher robustness, and instead reflects the quality of robustness transfer.

**Figure 1b.** Colored vertical bands indicate the same teacher architecture. For visual clarity, we apply a small horizontal jitter within each colored band, so left–right offsets inside a band are purely for visualization and have no meaning.

**Table 1.** All distillation results are obtained with RSLAD. Robust overfitting (RO) is defined as the drop from the best test PGD-20 accuracy during training to the final accuracy at epoch 200. With our schedule the peak typically occurs around epochs 100–110.

**Figure 3b.** For `Rebuffi2021Fixing`, we interpolate the teacher's soft distribution with the one-hot label as

$$\boldsymbol{t}_\alpha(\mathbf{x}) \;=\; (1-\alpha)\, f_T(\mathbf{x}) \;+\; \alpha\, \mathbf{e}_y, \quad \alpha \in [0,1], \tag{32}$$

where $f_T(\mathbf{x})$ is the teacher's probability vector and $\mathbf{e}_y$ is the one-hot vector for class $y$. When $\alpha = 0$ the target is the teacher distribution; when $\alpha = 1$ it is purely one-hot. Orange points correspond to different $\alpha$; numeric annotations indicate the one-hot proportion.

**Table 2.** CIFAR-10 with a ResNet-18 student distilled from `Bartoldson2024Adversarial` using RSLAD. The first 90 epochs are a warm-up on the full 50,000 examples. At epoch 90, we partition the training set into TAS and Non-TAS using equation 7; this split is then fixed for the remaining epochs.

**Table 5.** CIFAR-10 with a ResNet-18 student distilled from `Bartoldson2024Adversarial`. "Baseline" denotes plain RSLAD (no sample-wise weighting).

**Table 7.** CIFAR-10 with a ResNet-18 student distilled from the ERT `Chen2021LTD_WRN34_20`. Under this ERT, SAAD matches or slightly exceeds IGDM, which is consistent with our design goal: SAAD targets failure modes that arise under low transferability, and it does not aim to outperform existing methods when robustness transfer is already strong. Although one might hope to simply *pick an ERT* and avoid transferability issues, in practice it is rarely known in advance whether a given teacher will behave as an ERT or an IRT, and the teacher is often fixed (e.g., from a model zoo or an upstream system). SAAD therefore serves as a robust default: it consistently improves robustness when transferability is weak, while incurring no degradation when transferability is strong.

## C Additional Experiments

### C.1 Statistical Report

The main paper reports only mean results to preserve readability and avoid excessive font reduction in dense tables. Here, we provide full results, including standard deviations for three random seeds. See Table 12 for CIFAR-10 and Table 13 for CIFAR-100 and Tiny-ImageNet.

### C.2 Impact of Sample-wise Weighting Hyperparameters

As shown in Table 14, increasing the $\beta$ value enhances clean accuracy by placing greater emphasis on clean distillation. While small values of $\beta$ lead to modest improvements in clean accuracy with minimal reduction in adversarial robustness, overly large $\beta$ values cause substantial drops in both PGD and AutoAttack performance. This trade-off suggests that a moderate value offers the best balance between clean accuracy and robustness.

Table 12: Adversarial distillation results using two teacher and two student models on CIFAR-10. Clean, FGSM, PGD, C&W, and AA columns report accuracy (%) under each evaluation setting. Results are averaged over three random seeds with standard deviations.

| Model | Method | Bartoldson2024Adversarial | | | | | Gowal2021Improving | | | | |
|---|---|---|---|---|---|---|---|---|---|---|---|
| | | Clean | FGSM | PGD | C&W | AA | Clean | FGSM | PGD | C&W | AA |
| ResNet-18 | PGD-AT | $84.27_{\pm0.10}$ | $52.10_{\pm0.34}$ | $42.34_{\pm0.09}$ | $42.29_{\pm0.07}$ | $40.85_{\pm0.14}$ | $84.27_{\pm0.10}$ | $52.10_{\pm0.34}$ | $42.34_{\pm0.09}$ | $42.29_{\pm0.07}$ | $40.85_{\pm0.14}$ |
| | TRADES | $82.70_{\pm0.10}$ | $57.14_{\pm0.48}$ | $48.81_{\pm0.52}$ | $48.08_{\pm0.54}$ | $46.46_{\pm0.57}$ | $82.70_{\pm0.10}$ | $57.14_{\pm0.48}$ | $48.81_{\pm0.52}$ | $48.08_{\pm0.54}$ | $46.46_{\pm0.57}$ |
| | ARD | $84.63_{\pm0.15}$ | $56.57_{\pm0.45}$ | $44.48_{\pm0.26}$ | $43.44_{\pm0.45}$ | $41.66_{\pm0.53}$ | $84.39_{\pm0.28}$ | $52.47_{\pm0.35}$ | $42.18_{\pm0.32}$ | $42.22_{\pm0.21}$ | $40.79_{\pm0.40}$ |
| | IAD | $84.43_{\pm0.25}$ | $56.64_{\pm0.20}$ | $44.82_{\pm0.15}$ | $43.47_{\pm0.11}$ | $41.80_{\pm0.15}$ | $84.28_{\pm0.29}$ | $52.25_{\pm0.13}$ | $42.16_{\pm0.24}$ | $42.19_{\pm0.09}$ | $40.70_{\pm0.10}$ |
| | RSLAD | $84.28_{\pm0.11}$ | $57.20_{\pm0.62}$ | $47.17_{\pm0.33}$ | $46.07_{\pm0.42}$ | $44.42_{\pm0.34}$ | $83.83_{\pm0.36}$ | $51.59_{\pm0.44}$ | $42.03_{\pm0.31}$ | $42.22_{\pm0.28}$ | $40.57_{\pm0.38}$ |
| | AKD | $84.62_{\pm0.14}$ | $56.29_{\pm0.08}$ | $44.42_{\pm0.09}$ | $43.43_{\pm0.12}$ | $41.79_{\pm0.12}$ | $84.32_{\pm0.19}$ | $52.13_{\pm0.34}$ | $42.38_{\pm0.07}$ | $42.44_{\pm0.17}$ | $40.95_{\pm0.06}$ |
| | AdaAD | $\underline{85.07}_{\pm0.07}$ | $57.54_{\pm0.21}$ | $47.16_{\pm0.21}$ | $46.06_{\pm0.18}$ | $44.55_{\pm0.17}$ | $85.04_{\pm0.11}$ | $53.85_{\pm0.23}$ | $44.65_{\pm0.28}$ | $44.90_{\pm0.15}$ | $43.27_{\pm0.18}$ |
| | IGDM | $84.75_{\pm0.18}$ | $58.38_{\pm0.32}$ | $47.56_{\pm0.25}$ | $46.43_{\pm0.21}$ | $44.94_{\pm0.16}$ | $\underline{85.67}_{\pm0.22}$ | $58.14_{\pm0.44}$ | $48.58_{\pm0.18}$ | $46.98_{\pm0.40}$ | $44.76_{\pm0.52}$ |
| | **SAAD-C** | $\mathbf{85.54}_{\pm0.01}$ | $\mathbf{61.92}_{\pm0.11}$ | $\underline{53.18}_{\pm0.21}$ | $\underline{52.05}_{\pm0.30}$ | $\underline{50.14}_{\pm0.24}$ | $\mathbf{86.39}_{\pm0.01}$ | $\mathbf{60.55}_{\pm0.08}$ | $\underline{51.91}_{\pm0.45}$ | $\underline{52.06}_{\pm0.14}$ | $\underline{49.72}_{\pm0.16}$ |
| | **SAAD** | $84.27_{\pm0.18}$ | $\underline{61.44}_{\pm0.25}$ | $\mathbf{53.39}_{\pm0.23}$ | $\mathbf{52.39}_{\pm0.28}$ | $\mathbf{50.34}_{\pm0.08}$ | $83.69_{\pm0.18}$ | $\underline{59.74}_{\pm0.14}$ | $\mathbf{52.89}_{\pm0.40}$ | $\mathbf{52.36}_{\pm0.18}$ | $\mathbf{50.35}_{\pm0.22}$ |
| MobileNetV2 | PGD-AT | $83.52_{\pm0.19}$ | $54.92_{\pm0.24}$ | $44.90_{\pm0.43}$ | $44.29_{\pm0.18}$ | $41.54_{\pm0.22}$ | $83.52_{\pm0.19}$ | $54.92_{\pm0.24}$ | $44.90_{\pm0.43}$ | $44.29_{\pm0.18}$ | $41.54_{\pm0.22}$ |
| | TRADES | $81.79_{\pm0.46}$ | $56.50_{\pm0.19}$ | $49.90_{\pm0.07}$ | $47.54_{\pm0.26}$ | $46.50_{\pm0.14}$ | $81.79_{\pm0.46}$ | $56.50_{\pm0.19}$ | $49.90_{\pm0.07}$ | $47.54_{\pm0.26}$ | $46.50_{\pm0.14}$ |
| | ARD | $83.66_{\pm0.39}$ | $55.09_{\pm0.22}$ | $44.71_{\pm0.06}$ | $43.49_{\pm0.11}$ | $41.24_{\pm0.09}$ | $83.62_{\pm0.09}$ | $54.62_{\pm0.48}$ | $44.60_{\pm0.31}$ | $44.18_{\pm0.25}$ | $41.47_{\pm0.17}$ |
| | IAD | $83.85_{\pm0.27}$ | $55.69_{\pm0.10}$ | $44.98_{\pm0.10}$ | $43.62_{\pm0.04}$ | $41.35_{\pm0.05}$ | $83.63_{\pm0.12}$ | $54.81_{\pm0.17}$ | $44.73_{\pm0.13}$ | $44.19_{\pm0.17}$ | $41.51_{\pm0.03}$ |
| | RSLAD | $83.22_{\pm0.18}$ | $55.54_{\pm0.30}$ | $46.09_{\pm0.79}$ | $44.80_{\pm0.84}$ | $42.56_{\pm0.60}$ | $83.41_{\pm0.36}$ | $54.60_{\pm0.06}$ | $45.01_{\pm0.20}$ | $44.41_{\pm0.23}$ | $41.78_{\pm0.16}$ |
| | AKD | $83.60_{\pm0.42}$ | $55.13_{\pm0.21}$ | $44.31_{\pm0.11}$ | $43.29_{\pm0.21}$ | $41.03_{\pm0.21}$ | $83.54_{\pm0.03}$ | $54.79_{\pm0.11}$ | $44.83_{\pm0.12}$ | $44.19_{\pm0.14}$ | $41.55_{\pm0.05}$ |
| | AdaAD | $\underline{84.42}_{\pm0.12}$ | $56.38_{\pm0.23}$ | $46.16_{\pm0.07}$ | $44.99_{\pm0.12}$ | $43.01_{\pm0.11}$ | $\underline{84.37}_{\pm0.08}$ | $54.40_{\pm0.39}$ | $44.40_{\pm0.41}$ | $44.59_{\pm0.40}$ | $41.95_{\pm0.49}$ |
| | IGDM | $84.07_{\pm0.09}$ | $57.31_{\pm0.14}$ | $47.39_{\pm0.02}$ | $45.43_{\pm0.05}$ | $43.57_{\pm0.11}$ | $84.13_{\pm0.49}$ | $\underline{57.93}_{\pm0.10}$ | $48.70_{\pm0.33}$ | $47.43_{\pm0.12}$ | $44.83_{\pm0.19}$ |
| | **SAAD-C** | $\mathbf{85.16}_{\pm0.10}$ | $\mathbf{60.53}_{\pm0.18}$ | $\underline{52.72}_{\pm0.13}$ | $\underline{51.26}_{\pm0.20}$ | $\underline{49.34}_{\pm0.09}$ | $\mathbf{84.81}_{\pm0.23}$ | $\mathbf{58.17}_{\pm0.15}$ | $\underline{51.09}_{\pm0.35}$ | $\mathbf{50.45}_{\pm0.29}$ | $\underline{48.08}_{\pm0.23}$ |
| | **SAAD** | $82.04_{\pm0.04}$ | $\underline{59.48}_{\pm0.20}$ | $\mathbf{53.69}_{\pm0.19}$ | $\mathbf{51.68}_{\pm0.28}$ | $\mathbf{49.88}_{\pm0.16}$ | $80.60_{\pm0.23}$ | $56.85_{\pm0.02}$ | $\mathbf{51.78}_{\pm0.05}$ | $\underline{50.25}_{\pm0.05}$ | $\mathbf{48.29}_{\pm0.10}$ |

Table 13: Adversarial distillation results using ResNet-18 and PreActResNet-18 as student models for CIFAR-100 and Tiny-ImageNet, respectively, with the `Wang2023Better` teacher (WRN-70-16 for CIFAR-100 and WRN-28-10 for Tiny-ImageNet). Clean, FGSM, PGD, C&W, and AA columns report accuracy (%) under each evaluation setting. Results are averaged over three random seeds with standard deviations.

| Method | CIFAR-100 | | | | | Tiny-ImageNet | | | | |
|---|---|---|---|---|---|---|---|---|---|---|
| | Clean | FGSM | PGD | C&W | AA | Clean | FGSM | PGD | C&W | AA |
| PGD-AT | $56.17_{\pm0.20}$ | $24.74_{\pm0.16}$ | $19.65_{\pm0.18}$ | $19.79_{\pm0.20}$ | $18.66_{\pm0.18}$ | $45.71_{\pm0.33}$ | $15.75_{\pm0.05}$ | $11.67_{\pm0.43}$ | $11.82_{\pm0.45}$ | $10.91_{\pm0.40}$ |
| TRADES | $53.37_{\pm0.36}$ | $28.72_{\pm0.26}$ | $25.12_{\pm0.12}$ | $23.11_{\pm0.11}$ | $22.32_{\pm0.14}$ | $42.06_{\pm0.11}$ | $19.58_{\pm0.04}$ | $17.15_{\pm0.19}$ | $14.03_{\pm0.34}$ | $13.33_{\pm0.29}$ |
| ARD | $58.13_{\pm0.22}$ | $29.71_{\pm0.14}$ | $24.97_{\pm0.13}$ | $22.22_{\pm0.27}$ | $20.84_{\pm0.12}$ | $55.69_{\pm0.49}$ | $30.13_{\pm0.01}$ | $26.62_{\pm0.07}$ | $22.09_{\pm0.18}$ | $19.90_{\pm0.10}$ |
| IAD | $57.59_{\pm0.13}$ | $29.85_{\pm0.30}$ | $25.21_{\pm0.06}$ | $22.41_{\pm0.13}$ | $21.00_{\pm0.14}$ | $53.75_{\pm0.49}$ | $29.85_{\pm0.14}$ | $26.95_{\pm0.16}$ | $22.40_{\pm0.12}$ | $20.56_{\pm0.30}$ |
| RSLAD | $56.68_{\pm0.34}$ | $30.87_{\pm0.18}$ | $27.27_{\pm0.07}$ | $23.91_{\pm0.09}$ | $22.66_{\pm0.19}$ | $53.18_{\pm0.11}$ | $30.14_{\pm0.27}$ | $27.69_{\pm0.03}$ | $22.86_{\pm0.04}$ | $21.42_{\pm0.13}$ |
| AKD | $58.22_{\pm0.16}$ | $29.14_{\pm0.19}$ | $24.35_{\pm0.03}$ | $21.80_{\pm0.06}$ | $20.61_{\pm0.13}$ | $54.48_{\pm0.37}$ | $27.62_{\pm0.04}$ | $23.59_{\pm0.33}$ | $19.56_{\pm0.16}$ | $17.73_{\pm0.12}$ |
| AdaAD | $58.57_{\pm0.49}$ | $31.72_{\pm0.04}$ | $28.00_{\pm0.10}$ | $24.40_{\pm0.48}$ | $23.15_{\pm0.30}$ | $\underline{57.26}_{\pm0.11}$ | $31.81_{\pm0.18}$ | $28.80_{\pm0.15}$ | $23.64_{\pm0.09}$ | $22.11_{\pm0.05}$ |
| IGDM | $56.36_{\pm0.32}$ | $32.95_{\pm0.13}$ | $29.68_{\pm0.08}$ | $25.91_{\pm0.12}$ | $24.81_{\pm0.28}$ | $57.15_{\pm0.08}$ | $31.98_{\pm0.18}$ | $29.02_{\pm0.17}$ | $23.94_{\pm0.04}$ | $22.52_{\pm0.11}$ |
| **SAAD-C** | $\mathbf{59.57}_{\pm0.66}$ | $\mathbf{36.05}_{\pm0.38}$ | $\underline{32.52}_{\pm0.31}$ | $\underline{28.72}_{\pm0.38}$ | $\underline{27.21}_{\pm0.34}$ | $\mathbf{57.33}_{\pm0.16}$ | $\underline{33.06}_{\pm0.38}$ | $\underline{29.62}_{\pm0.23}$ | $\underline{24.16}_{\pm0.32}$ | $\underline{22.69}_{\pm0.35}$ |
| **SAAD** | $\underline{59.11}_{\pm0.28}$ | $\underline{36.01}_{\pm0.13}$ | $\mathbf{32.71}_{\pm0.21}$ | $\mathbf{29.36}_{\pm0.16}$ | $\mathbf{27.58}_{\pm0.16}$ | $57.16_{\pm0.36}$ | $\mathbf{33.26}_{\pm0.32}$ | $\mathbf{29.95}_{\pm0.13}$ | $\mathbf{24.87}_{\pm0.29}$ | $\mathbf{23.42}_{\pm0.23}$ |

To explicitly demonstrate the superiority of this trade-off mechanism, we compare SAAD-C against an unweighted clean distillation baseline. Many existing adversarial distillation baselines (Goldblum et al., 2020; Zi et al., 2021; Huang et al., 2023) conceptually incorporate a clean distillation term in their objective:

$$\mathcal{L}_{\text{baseline}} = \mathcal{L}_{\text{AD}} + \frac{1}{N} \sum_{i=1}^{N} \beta \cdot \text{KL}(f_T(\mathbf{x}_i) \parallel f_S(\mathbf{x}_i)) \tag{33}$$

However, prior works typically disable this term (i.e., setting $\beta = 0$) because uniformly increasing $\beta$ results in an unfavorable trade-off, where the loss in robustness outweighs the gains in clean accuracy. To address this limitation, SAAD-C selectively applies the clean distillation term as defined in Equation 9. By introducing the $(1 - \tilde{w}_i)$ weight, SAAD-C restricts the clean distillation constraint mainly to non-TAS samples. To explicitly demonstrate this difference, we added an experiment sweeping $\beta$ for the unweighted baseline (denoted as 'SAAD + Clean KD') and compared against SAAD-C in Figure 5. While the SAAD + Clean KD baseline experiences a robustness drop to gain clean accuracy, a better clean-robustness trade-off can be observed in

Table 14: Effect of $\beta$ on CIFAR-10 with the `Gowal2021Improving` teacher and ResNet-18 student.

| $\beta$ | Clean | FGSM | PGD | C&W | AA |
|---|---|---|---|---|---|
| 0 | 83.69 | 59.74 | 52.45 | 52.36 | 50.35 |
| 0.05 | 84.72 | 60.13 | 52.77 | 52.32 | 50.19 |
| 0.1 | 85.49 | 59.70 | 52.45 | 52.18 | 50.14 |
| 0.15 | 85.91 | 60.21 | 52.45 | 52.21 | 49.90 |
| 0.2 | 86.39 | 60.55 | 51.91 | 52.06 | 49.72 |
| 0.25 | 87.93 | 59.40 | 49.18 | 49.36 | 47.02 |
| 0.3 | 88.13 | 57.79 | 44.88 | 45.17 | 42.65 |
| 0.5 | 88.19 | 56.44 | 42.65 | 43.06 | 40.78 |

Table 15: AD with SWA results on CIFAR-10 with the `Gowal2021Improving` teacher and ResNet-18 student.

| Method | Clean | FGSM | PGD | C&W | AA |
|---|---|---|---|---|---|
| ARD | 85.33 | 58.06 | 48.69 | 47.68 | 46.12 |
| IAD | 85.01 | 58.39 | 48.94 | 47.78 | 46.10 |
| RSLAD | 84.74 | 59.96 | 49.40 | 48.16 | 46.60 |
| AKD | 85.20 | 57.95 | 48.51 | 47.44 | 46.04 |
| AdaAD | 86.11 | 56.70 | 48.08 | 48.05 | 46.19 |
| IGDM | 86.09 | 58.85 | 50.13 | 49.83 | 48.18 |
| SAAD-C | **86.39** | **60.55** | 51.91 | 52.06 | 49.72 |
| SAAD | 83.69 | 59.74 | **52.89** | **52.36** | **50.35** |

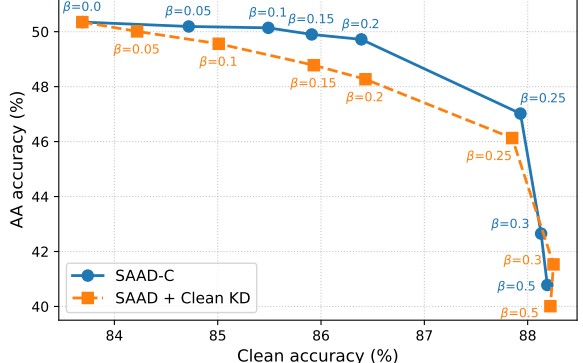

Figure 5: Clean vs. AutoAttack accuracy Pareto curve across varying $\beta$ on CIFAR-10 with the `Gowal2021Improving` teacher and ResNet-18 student. SAAD-C achieves a more favorable trade-off than the unweighted clean distillation baseline.

Figure 6: Correlation between TAS ratio and AA accuracy across varying PGD inner steps and perturbation bounds ($\epsilon$). The linear trend suggests that TAS ratio is closely associated with robustness in this diagnostic sweep.

SAAD-C thanks to the sample-wise weighting mechanism. This confirms that SAAD-C effectively recovers clean accuracy without severely compromising adversarial robustness, unlike standard clean distillation.

## C.3 Effect of Attack Strength on Transferability and Robustness

To investigate the impact of attack strength on robustness transfer, we conducted two sets of experiments on CIFAR-10 with a ResNet-18 student and the `Gowal2021Improving` teacher. Specifically, we varied the number of PGD inner steps and the maximum perturbation bound $\epsilon$ during student training. As shown in Figure 6, increasing the attack strength consistently elevates the TAS ratio. Furthermore, we observe a linear correlation between the TAS ratio and the final AutoAttack accuracy. This suggests that improved transferability of student-crafted adversarial examples is closely associated with stronger student robustness in this setting.

## C.4 SWA analysis

To ensure consistency, we apply the SWA technique across all adversarial distillation baselines and evaluate their performance under identical conditions. As shown in Table 15, applying SWA generally improves the robustness of existing AD methods to some extent. Nevertheless, both SAAD and SAAD-C consistently outperform all baselines, achieving the highest robustness. This indicates that the observed gains are not merely due to SWA but rather attributable to the design of our proposed distillation framework.

Table 16: Compatibility of SAAD weighting with other AD methods. We report test accuracy (%) on CIFAR-10 with ResNet-18 student distilled from the `Gowal2021Improving` teacher.

| Method | Clean | FGSM | PGD | C&W | AA |
|---|---|---|---|---|---|
| ARD | 84.39 | 52.47 | 42.18 | 42.22 | 40.79 |
| ARD + SAAD weighting | 81.95 | 56.48 | 50.04 | 50.39 | 48.09 |
| RSLAD | 83.83 | 51.59 | 42.03 | 42.22 | 40.57 |
| RSLAD + SAAD weighting | 81.74 | 57.03 | 50.54 | 50.46 | 48.53 |
| AdaAD | 85.04 | 53.85 | 44.65 | 44.90 | 43.27 |
| AdaAD + SAAD weighting | 84.16 | 58.93 | 52.16 | 51.99 | 49.97 |
| IGDM | 85.67 | 58.14 | 48.58 | 46.98 | 44.76 |
| SAAD-C | 86.39 | 60.55 | 51.91 | 52.06 | 49.72 |
| SAAD | 83.69 | 59.74 | 52.89 | 52.36 | 50.35 |

Table 17: Throughput (epochs/hour) and peak GPU memory (MB) on CIFAR-10.

| Method | Epochs/hour | Memory (MB) |
|---|---|---|
| ARD | 17.76 | 4670 |
| IAD | 10.93 | 4670 |
| RSLAD | 10.93 | 4670 |
| AKD | 17.85 | 4670 |
| AdaAD | 1.23 | 26795 |
| IGDM | 1.17 | 26795 |
| **SAAD-C** | 4.73 | 26301 |
| **SAAD** | 4.83 | 26301 |

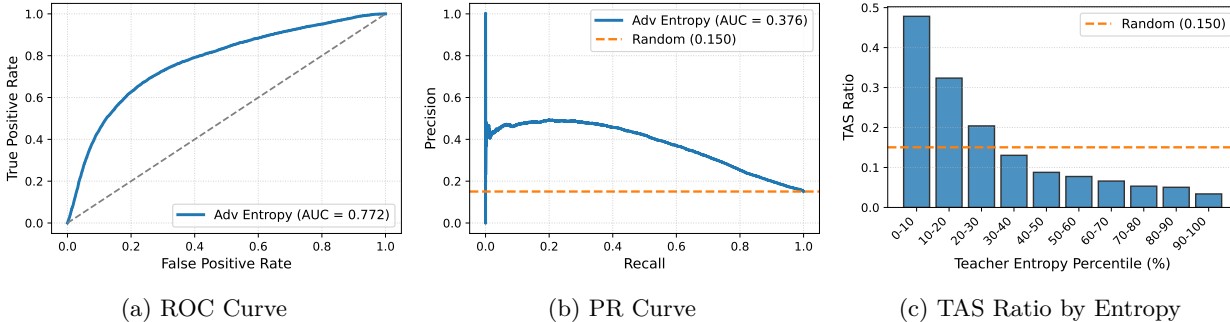

(a) ROC Curve      (b) PR Curve      (c) TAS Ratio by Entropy

Figure 7: Quantitative evaluation of the entropy proxy on CIFAR-10 using a ResNet-18 student and the `Gowal2021Improving` teacher. The metrics show that the teacher's entropy correlates with transferability, with non-TAS samples predominantly located in low-entropy regimes.

## C.5 Compatibility of SAAD Weighting with Other AD Methods

SAAD's entropy-based weighting scheme can be seamlessly integrated into the outer minimization of existing AD methods, as it does not require any modification to their inner optimization or loss formulation. To verify this, we apply the SAAD weighting design on top of ARD, RSLAD, and AdaAD. All experiments follow the same setting in the main paper, i.e., distillation on CIFAR-10 with a ResNet-18 student and the `Gowal2021Improving` teacher. In Table 16, applying SAAD weighting to existing methods consistently improves robustness across all attacks. However, these variants remain slightly less effective than the full SAAD method.

## C.6 Computational Resource

Table 17 reports throughput and peak memory on an NVIDIA A6000 (Ubuntu, Python 3.8, PyTorch). The larger memory usage for AdaAD/IGDM/SAAD is primarily due to backpropagating through the teacher during the inner maximization: IGDM and AdaAD run iterative inner loops with multiple teacher backward passes, whereas SAAD replaces this loop with a first-order approximation requiring only a single backward pass, yielding almost four times speedup while keeping peak memory comparable. By contrast, lighter baselines (ARD/AKD/RSLAD) avoid teacher backpropagation in the inner loop and therefore use much less memory and train faster. Importantly, the "without incurring additional computational cost" claim in the main text refers to SAAD's entropy-based weighting itself. Applied to gradient-free on teacher for inner-maximization methods such as ARD and RSLAD, the weighting improves robustness (see Table 16); the weighting does not increase memory or computation.

Table 18: Comparison of clean and adversarial entropy for sample-wise weighting on CIFAR-10 using a ResNet-18 student.

| Teacher | Weighting Metric | Clean | FGSM | PGD | C&W | AA |
|---|---|---|---|---|---|---|
| Bartoldson2024Adversarial | Clean Entropy | 80.67 | 57.40 | 51.85 | 49.54 | 48.11 |
| | SAAD | 84.27 | 61.44 | 53.39 | 52.39 | 50.34 |
| Gowal2021Improving | Clean Entropy | 80.78 | 56.49 | 50.77 | 49.87 | 47.85 |
| | SAAD | 83.69 | 59.74 | 52.89 | 52.36 | 50.35 |

### C.7 Quantifying the Entropy Proxy

We evaluate the teacher's entropy $H(f_T(\mathbf{x} + \boldsymbol{\delta}_S))$ as a proxy for adversarial transferability on the CIFAR-10 training set, employing a ResNet-18 student and the Gowal2021Improving teacher. As shown in Figure 7, the ROC curve yields an AUC of 0.772. The PR curve outperforms the random baseline. The distribution across entropy percentiles further indicates that non-transferable samples are predominantly concentrated in the lowest entropy regimes. These results suggest that the entropy proxy provides a practical continuous signal to identify and down-weight the overconfident predictions.

### C.8 Comparison of Clean and Adversarial Entropy

To validate the use of adversarial entropy $H(f_T(\mathbf{x} + \boldsymbol{\delta}_S))$ over clean entropy $H(f_T(\mathbf{x}))$ for sample-wise weighting, we evaluate both metrics using a ResNet-18 student. Table 18 presents the robustness outcomes on CIFAR-10. Clean entropy solely measures static uncertainty on unperturbed images, failing to capture the dynamic transferability of student-crafted perturbations and rendering it a sub-optimal proxy. In contrast, adversarial entropy explicitly evaluates the teacher's response to the attack. By integrating this dynamic measure, SAAD achieves superior robustness across all evaluation metrics.

### C.9 Impact of Underconfident Soft Labels in Adversarial Distillation.

One potential concern is that underconfident (high-entropy) soft labels may provide noisy supervision and thus harm adversarial distillation. To evaluate this, we analyze the teacher's prediction reliability in high-entropy regions. For each teacher, we distill a student using RSLAD on CIFAR-10. After training, we generate 10-step PGD adversarial examples on the student from the training set and compute the entropy of the teacher's output on these perturbed inputs. We then sort the samples by teacher-output entropy on student-generated adversarial inputs (transfer attacks) and report the teacher's prediction accuracy (transfer adversarial accuracy) on the top 20%, 40%, 60%, and 80% highest-entropy subsets. As shown in Table 19, transfer adversarial accuracy remains consistently above 94% even in high-entropy regions, suggesting that underconfident soft labels largely provide stable supervision rather than introducing significant noise. These results support the validity of entropy-based weighting: low-entropy, overconfident predictions are more indicative of high-variance, overfitting-prone samples, whereas high-entropy samples help stabilize training by reducing adversarial variance and enabling more effective knowledge transfer.

While transfer adversarial accuracy is high, it does not reach 100%, implying that a small fraction of teacher predictions deviate from the ground truth. To investigate whether explicitly correcting such deviations yields further benefits, we incorporate the Error-Corrective Label Swapping (ELS) technique proposed in DGAD (Park & Min, 2024), which swaps the true-label and max-logit probabilities for misclassified cases on both clean and adversarial examples. We apply ELS on top of SAAD and evaluate the resulting student on CIFAR-10 with a ResNet-18 student distilled from the Gowal2021Improving teacher. The results in Table 20 show that applying ELS yields only marginal changes across all evaluation metrics. While ELS explicitly corrects teacher outputs in misclassified cases, the gains remain negligible. This indicates that underconfident soft labels, despite their uncertainty, do not substantially degrade supervision, and explicit correction offers limited practical benefit. This confirms that the teacher's uncertainty signals are inherently

Table 19: Transfer adversarial accuracy (%) of teachers on high-entropy subsets. Even for high-entropy samples, teacher accuracy remains very high, indicating that underconfident labels still provide reliable supervision.

| Teacher | Top 20% | Top 40% | Top 60% | Top 80% | Average |
|---|---|---|---|---|---|
| Teacher A[†] | 94.50 | 97.06 | 97.99 | 98.46 | 98.73 |
| Teacher B[‡] | 97.06 | 98.50 | 99.00 | 99.24 | 99.39 |

[†]`Bartoldson2024Adversarial`, [‡]`Gowal2021Improving`

Table 20: Impact of ELS on SAAD. ELS leads to only minor changes across all metrics, suggesting that correcting underconfident predictions provides limited additional benefit.

| Method | Clean | FGSM | PGD | C&W | AA |
|---|---|---|---|---|---|
| SAAD | 83.69 | 59.74 | 52.89 | 52.36 | 50.35 |
| + ELS | 83.86 | 59.78 | 52.75 | 52.22 | 50.07 |

Table 21: Sensitivity analysis of adversarial variance (AVar) estimation across different $K$ and $N$ configurations on CIFAR-10 using a ResNet-18 student.

| Teacher | Default ($K=2, N=2$) | $K=3$ | $K=5$ | $N=3$ | $N=4$ | $N=5$ |
|---|---|---|---|---|---|---|
| `Rebuffi2021Fixing` | 0.0267 | 0.0273 | 0.0275 | 0.0441 | 0.0556 | 0.0694 |
| `Chen2021LTD` | 0.0059 | 0.0061 | 0.0060 | 0.0100 | 0.0140 | 0.0168 |
| `Bartoldson2024Adversarial` | 0.0834 | 0.0817 | 0.0805 | 0.1152 | 0.1405 | 0.1588 |
| `Gowal2021Improving` | 0.3058 | 0.3016 | 0.3061 | 0.4495 | 0.5386 | 0.6448 |

informative rather than noisy, validating our approach of leveraging raw soft labels directly through entropy weighting.

### C.10 Sensitivity Analysis of Adversarial Variance Estimation

To qualify the reliability of our adversarial variance (AVar) estimation, we conduct a sensitivity analysis on CIFAR-10 using a ResNet-18 student across four robust teachers. We systematically vary the key estimation hyperparameters: the number of repetitions $K \in \{2, 3, 5\}$ and the number of splits $N \in \{2, 3, 4, 5\}$. Table 21 presents the estimated AVar values under these configurations. Increasing the number of repetitions $K$ yields negligible differences, confirming that the estimation remains statistically stable across random splits. Conversely, increasing the number of splits $N$ leads to an increase in the absolute AVar values, which is a natural consequence of reducing the subset size $|D_j^{(k)}|$ available for the adversarial training phase. Notably, the relative ordering of AVar among the four teachers remains strictly consistent across all configurations of $K$ and $N$. This invariant ordering demonstrates that the estimation method reliably captures the comparative variance characteristics of different teachers.

## D Limitations and Future Work

**Limitations** Our experiments primarily evaluate CIFAR-10, CIFAR-100, and Tiny-ImageNet using standard robust teachers; scaling these findings to larger datasets such as ImageNet requires further investigation. Moreover, while we identify the properties that induce ineffective adversarial distillation, the fundamental mechanism explaining why high-capacity robust models naturally develop IRT behavior remains an open question. Finally, the theoretical lower bound established in Lemma 2, which links TAS and teacher uncertainty, can become vacuous if the minimum class probability approaches zero, causing the logarithmic term to diverge. Extending this analysis into a comprehensive formal framework with broader theoretical guarantees remains a vital open challenge.

**Discussion** Approaching this formal challenge is non-trivial, as a general theoretical understanding of adversarial training remains limited due to the non-convex min–max objective; thus, most analyses focus on simplified settings (Charles et al., 2019; Li et al., 2020; Zou et al., 2021; Javanmard & Soltanolkotabi, 2022; Chen et al., 2023; Tanner et al., 2024). While our study is primarily empirical, these frameworks offer a valuable interpretive lens. Specifically, works on margin geometry and trade-offs (Tsipras et al., 2019; Javanmard & Soltanolkotabi, 2022; Tanner et al., 2024) could explain why highly robust teachers

produce qualitatively different, overconfident output distributions that potentially dictate TAS concentration. Furthermore, recent theoretical work highlights that robust learning under adversarial conditions can remain challenging even in stylized regimes (Balcan et al., 2023) and is not automatically resolved by scaling model capacity (Vilucchio et al., 2025), aligning with our observation that capacity gaps alone cannot fully explain distillation failures. Finally, the non-robust features framework (Ilyas et al., 2019) helps clarify when student-crafted perturbations will or will not fool the teacher, suggesting non-TAS samples might naturally emerge when students leverage predictive yet teacher-misaligned features.

**Future Work** Motivated by these theoretical connections, a critical research direction is to transition from empirical diagnostics to a fundamental causal understanding of adversarial distillation. Specifically, identifying the architectural, optimization, and data-regime conditions under which robust teachers emerge as ERTs or IRTs is essential. As suggested by Allen-Zhu & Li (2023) and Li & Li (2025a;b), adopting a feature-learning perspective is particularly promising for resolving these questions. Under this perspective, we hypothesize at the sample level that a small subset of intrinsically hard training examples elicits high-entropy, noise-like guidance from ERTs, which the student effectively ignores. Conversely, IRTs enforce overly confident supervision on complex features beyond the representational capacity of the student. This capacity mismatch compels the student to memorize spurious noise to match the teacher's targets, providing vulnerable attack directions that ultimately drive robust overfitting.

