# OpenReview forum: "Sample-wise Adaptive Weighting for Transfer Consistency in Adversarial Distillation"
_TMLR — Accepted by TMLR_

### Review · Reviewer_vBzq · 2026-03-03

**Summary Of Contributions:**

This paper studies why adversarial distillation (AD) can saturate, where stronger teachers do not consistently yield more robust students. It argues that the main driver is student to teacher attack transferability, and proposes the TAS ratio to quantify this effect. The method, **SAAD**, applies sample-wise weights during AD using the teacher’s entropy on student-adversarial inputs as a proxy for transfer consistency. It also introduces **SAAD-C** to tune the robustness–clean accuracy trade-off. Across multiple datasets, students, and teachers, SAAD improves robustness (including AutoAttack) over prior AD baselines. The results also suggest that low transferability is associated with higher adversarial variance and robust overfitting, and that the proposed weighting helps mitigate this behavior.

**Audience:**

Yes

**Audience Explanation:**

The paper targets an audience that cares about robustness, adversarial training/distillation, and why AD saturates.

**Broader Impact Concerns:**

This work can improve the robustness and reliability of distilled models under adversarial perturbations, which is relevant for safety-critical and security-sensitive deployments. A potential risk is dual-use: better understanding of transferability and failure modes could also inform stronger attacks or attack selection strategies.

**Claims And Evidence:**

Yes

**Claims Explanation:**

### Overall claims are moderately backed by evidence.

**Claim 1:** We identify adversarial transferability as a key factor for effective adversarial distillation, explaining why stronger teachers can fail to improve student robustness.
**Evidence:** Moderate. The paper shows transferability (TAS) tracks robustness outcomes and supports the story with variance/overfitting analysis.
**Gap:** The evidence is mostly correlational. To fully justify “explains why,” it would help to add intervention-style tests that change transferability while keeping teacher/student fixed.


**Claim 2:** We propose Sample-wise Adaptive Adversarial Distillation (SAAD), which selectively emphasizes transferable samples to mitigate high-variance effects and improve robustness.
**Evidence:** Mostly. Robustness gains are well supported, and the variance-mitigation narrative is consistent with the reported trends.
**Gap:** The “selectively emphasizes transferable samples” part relies on entropy as a proxy for TAS. This needs explicit quantification (e.g., entropy predicting TAS labels) and a comparison to simpler confidence-weighting baselines.


**Claim 3:** We show that our method consistently improves both robustness and clean accuracy across diverse settings, outperforming prior adversarial distillation approaches.
**Evidence:** Strong for robustness; mixed for clean accuracy. Robustness improvements are consistent across settings and evaluations.
**Gap:** Clean accuracy gains appear mainly via SAAD-C’s trade-off rather than universally. The claim should be qualified or backed with a clear clean-vs-robustness Pareto curve across the trade-off parameter.

**Requested Changes:**

## Requested Changes (Experiments)

1) **Quantify entropy as a proxy for transferability.**
Compute the TAS label on a subset of samples, then report how well teacher entropy on student-adversarial inputs predicts TAS (AUC/PR-AUC, and a simple bin plot).

2) **Clean vs adversarial entropy ablation.**
Directly compare weighting $(\(H(f_T(x))\) versus \(H(f_T(x+\delta_S))\))$ to show the gain is specific to student→teacher transfer, not just generic uncertainty.

3) **Intervention-style transferability sweep.**
Vary the student attack strength used during training or analysis (e.g., $\(\epsilon\)$ or number of steps), report TAS ratio vs attack strength, and show how student robustness changes in the same sweep.

4) **Clarify the robustness vs clean accuracy claim.**
Include a clean-vs-robustness trade-off curve for SAAD-C (varying the trade-off parameter) and rephrase the main text to reflect that clean gains come from this tunable trade-off.

---

> ### Author Response · Authors · 2026-03-23
> **Reply to Reviewer vBzq**
>
> **1. Quantify entropy as a proxy for transferability.**
>
> **A.** We computed the true transferability labels on the CIFAR-10 training set and quantified the predictive power of our proxy. ROC curve, PR curve, and bin plot have been added as **Figure 7 in Section C.7** of the revised manuscript. Specifically, we evaluated the teacher's entropy employing a ResNet-18 student and the Gowal2021Improving teacher. The empirical evaluation yields an ROC curve with an AUC of 0.772. Given the inherent class imbalance, the PR curve achieves an AUC of 0.376, which outperforms the 0.150 random baseline. Furthermore, the distribution across entropy percentiles indicates that non-transferable samples are predominantly concentrated in the lowest entropy regimes. These results confirm that the entropy proxy provides a practical continuous signal to identify and down-weight overconfident predictions.
>
> **2. Clean vs. adversarial entropy ablation.**
>
> **A.** As shown in the table below, using clean entropy for weighting is not optimal. Clean entropy only measures the teacher's static uncertainty on original images and misses the dynamic transferability of the student-crafted perturbations. In contrast, SAAD explicitly captures how the teacher reacts to the attack, leading to superior performance. We have incorporated this analysis into **Section C.8 (Table 18)** of the revised manuscript.
>
> | Teacher | Weighting Metric | Clean | FGSM | PGD | C&W | AA |
> | :--- | :--- | :---: | :---: | :---: | :---: | :---: |
> | Bartoldson2024Adversarial| Clean Entropy | 80.67 | 57.40 | 51.85 | 49.54 | 48.11 |
> | Bartoldson2024Adversarial| SAAD| 84.27 | 61.44 | 53.39 | 52.39 | 50.34 |
> | Gowal2021Improving | Clean Entropy | 80.78 | 56.49 | 50.77 | 49.87 | 47.85 |
> | Gowal2021Improving| SAAD| 83.69| 59.74| 52.89| 52.36| 50.35|
>
>
> **3. Intervention-style transferability sweep.**
>
> To explicitly observe how the student's robustness responds to changes in adversarial transferability, we conducted the suggested intervention-style transferability sweep. Please refer to **Section C.3 and Figure 6** for the detailed correlation plots and discussion. Specifically, we varied the student's training attack strength via PGD inner steps and perturbation bound $\epsilon$ on CIFAR-10 with ResNet-18 student and Gowal2021Improving teacher. Across both sweeps, increasing the attack strength consistently elevated the TAS ratio, which exhibited a linear correlation with the final AutoAttack accuracy. This empirically validates that adversarial transferability drives the proportional improvement in student robustness.
>
>
> **4. Clarify the robustness vs clean accuracy claim.**
>
> We agree and clarify that the improvements in clean accuracy come from a tunable trade-off. Existing adversarial distillation baselines conceptually incorporate a clean distillation term in their objective:
>
> $$\mathcal{L}\_{\mathrm{baseline}} = \mathcal{L}\_{\mathrm{AD}} + \frac{1}{N} \sum_{i=1}^{N} \beta \cdot \mathrm{KL}(f\_T(x\_i) \parallel f\_S(x\_i))$$
>
> However, prior works typically disable this term because increasing $\beta$ results in an unfavorable trade-off, where the loss in robustness outweighs the gains in clean accuracy.
> To address this, SAAD-C selectively applies the clean distillation term. As defined in Eq. 9:
>
> $$\mathcal{L}\_{\mathrm{SAAD-C}} = \mathcal{L}\_{\mathrm{SAAD}} + \frac{1}{N} \sum\_{i=1}^{N} \beta \cdot (1 - \tilde{w}\_i) \cdot \mathrm{KL}(f\_T(x\_i) \parallel f\_S(x\_i))$$
>
> By introducing the $(1 - \tilde{w}_i)$ weight, SAAD-C restricts the clean distillation constraint mainly to low-transferability (non-TAS) samples. To explicitly demonstrate this difference, we added an experiment sweeping $\beta$ for an unweighted baseline (SAAD + Clean KD) and compared it against SAAD-C. The results for SAAD-C are taken directly from Table 14 in our manuscript. As shown in the table below, the unweighted baseline experiences a larger robustness drop to gain clean accuracy (e.g., -2.08% AA at $\beta = 0.20$). In contrast, SAAD-C achieves a more favorable trade-off (+2.70% Clean, -0.63% AA at $\beta = 0.20$).
>
> | Parameter ($\beta$) | SAAD-C (Clean) | SAAD-C (AA) | SAAD + Clean KD (Clean) | SAAD + Clean KD (AA) |
> | :---: | :---: | :---: | :---: | :---: |
> | 0.00 | 83.69 | 50.35 | 83.69 | 50.35 |
> | 0.05 | 84.72 | 50.19 | 84.22 | 50.01 |
> | 0.10 | 85.49 | 50.14 | 85.01 | 49.56 |
> | 0.15 | 85.91 | 49.90 | 85.93 | 48.78 |
> | 0.20 | 86.39 | 49.72 | 86.43 | 48.27 |
> | 0.25 | 87.93 | 47.02 | 87.85 | 46.13 |
> | 0.30 | 88.13 | 42.65 | 88.25 | 41.53 |
> | 0.50 | 88.19 | 40.78 | 88.22 | 40.01 |
>
> Following your suggestion, we have revised our claims throughout the manuscript—specifically in the Introduction, Experimental Results, and Conclusion—to clarify that our method provides a tunable clean-robustness trade-off rather than a universal improvement in both metrics. Furthermore, we have added the corresponding Pareto curve as **Figure 5 in Section C.2** to explicitly visualize this trade-off.

---

### Review · Reviewer_3qim · 2026-03-06

**Summary Of Contributions:**

The paper investigates the phenomenon of *robust saturation* in adversarial distillation (AD), when employing stronger robust teacher models does not necessarily yield more robust student models.
The authors show that the standard explanation based on teacher–student capacity gaps is incomplete.
Instead, they identify the fraction of student-crafted adversarial examples that remain effective against the teacher to be the factor determining whether a teacher is effective or ineffective for robustness transfer. They call this propriety adversarial transferability.

Given this setting the presentation and analysis consists of three distinct parts. First, the authors show empirically that IRTs produce overconfident (low-entropy) outputs on student-generated adversarial inputs, while ERTs maintain broader entropy distributions.
Second, they extend the bias–variance decomposition to adversarial distillation, showing that overconfident teacher logits correlate with high adversarial variance in the student, which in turn correlates with robust overfitting.
Third, they partition training samples into transferable (TAS) and non-transferable subsets based on a KL-divergence criterion (Eq. 7), and show that non-TAS samples concentrate overconfident supervision and drive overfitting.

Based on these observations, the authors propose the main method of the paper Sample-wise Adaptive Adversarial Distillation (SAAD), which reweights the distillation loss by the entropy of the teacher's output on student-perturbed inputs. Experiments on CIFAR-10, CIFAR-100, and Tiny-ImageNet show AutoAttack improvements over prior AD methods.

**Additional Comments:**

The paper is a well-executed empirical study with a clear arc from diagnosis to solution. The core insight is valuable and the method is effective. My main reservations are about the gap between the formal justification (Lemma 2) and the empirical success of the entropy proxy, and the reliability of the variance estimates. Addressing these (even through honest discussion of limitations rather than new experiments) would make this a solid TMLR contribution.

**Audience:**

Yes

**Audience Explanation:**

Adversarial distillation is at the intersection of adversarial robustness and model compression, both of which are actively studied in the TMLR community.
The finding that state-of-the-art robust teachers from RobustBench can actually *degrade* student robustness under existing AD methods is surprising and practically relevant.

More broadly, the observation that highly robust models can produce overconfident, low-entropy outputs on transferred adversarial examples touches on ongoing discussions about robustness, calibration, and the geometry of adversarial perturbations.
However, the paper does not really engage with this broader theoretical context.
The related work covers the AD literature well, but the theoretical understanding of why adversarial examples transfer between models - and under what conditions transferability concentrates or fails - has advanced considerably (e.g., the non-robust features framework of Ilyas et al., geometric analyses such as Javanmard and Soltanolkotabi, Tanner et al.) and is directly relevant to the paper's thesis. Connecting to this literature would strengthen the contribution.

**Broader Impact Concerns:**

The paper does not raise significant broader impact concerns. Improving adversarial robustness transfer to compact models has positive implications for deploying robust models in resource-constrained settings.

**Claims And Evidence:**

Yes

**Claims Explanation:**

The paper's main empirical claim is well supported. Figure 1c shows a clear positive trend between TAS ratio and student AutoAttack accuracy across teachers and methods, and this is substantially more convincing than the corresponding plots for robustness (Figure 1a) or architecture size (Figure 1b).
The diagnostic analysis in Section 3 is one of the paper's strengths: the entropy histograms (Figure 2), the label-mixing interpolation experiment (Figure 3b), and the TAS/non-TAS partition experiments (Figure 4) build a coherent narrative. The label-mixing experiment is particularly nice as it provides a controlled intervention rather than just a correlation.

On the experimental side, the improvements are consistent across datasets, student architectures (ResNet-18, MobileNetV2, PreActResNet-18), and diverse evaluation protocols including OODRobustBench and black-box attacks (Table 8). Results are averaged over three seeds with standard deviations in the appendix. The method itself is simple and introduces no additional computational overhead, which is a practical plus, and the fast inner maximization via first-order Taylor approximation (Algorithm 2) is a useful contribution on its own.

That said, there are two aspects where the evidence is weaker. The adversarial variance estimates (AVar) rely on training only $N=2$ students on half the data with $K=2$ repetitions (Algorithm 1, Section A.2). Computing a KL-based variance from just 2 models per repetition is inherently noisy, and the AVar values in Table 1 and Figure 3 should be interpreted with caution. It would help to see confidence intervals, or at least a discussion of why the relative ordering of AVar across teachers is stable given this estimation noise.

The second concern is Lemma 2, the formal justification for using entropy as a proxy for TAS. The bound $\text{TAS}(x) \geq H(f_T(x+\delta_S)) + \log m$ depends on $m = \min_i q_i$, the smallest probability in the teacher's own adversarial output $f_T(x+\delta_T)$. This quantity is never computed or discussed. For the IRTs that the paper specifically targets $m$ could be very small, pushing $\log m$ toward $-\infty$ and making the bound vacuous. The method works well in practice, so entropy clearly captures something useful, but the formal bound does not fully explain why. This gap should be acknowledged honestly.

**Requested Changes:**

[Required] **Discuss the tightness and practical relevance of Lemma 2.** The lower bound involves $\log m = \log(\min_i q_i)$, where $q = f_T(x + \delta_T)$ as explained before. The authors should either (a) provide empirical estimates of $m$ across teachers, showing the bound is non-trivial for a reasonable fraction of samples, or (b) provide a tighter analysis. At minimum, the limitation should be stated explicitly and the entropy proxy framed as an empirically motivated heuristic rather than a formally justified choice.

[Required] **Strengthen the adversarial variance estimation or qualify its reliability.** With $N=2$ splits and $K=2$ repetitions, each variance estimate comes from just 2 models per run.  Either increase $N$ and $K$ and report confidence intervals, or provide a clear argument for why the observed trends are robust to estimation noise.

[Required] **Engage more with the theoretical adversarial robustness literature.** The paper treats adversarial transferability as a purely empirical phenomenon, but there is substantial theoretical work on why adversarial examples transfer, on the accuracy–robustness tradeoff, and on the geometry of adversarially trained models that is directly relevant. Specifically:

- Ilyas et al., _"Adversarial examples are not bugs, they are features_ , the non-robust features framework helps explain when student-crafted perturbations will or will not fool the teacher.
- Tsipras et al., _"Robustness may be at odds with accuracy"_ relevant to understanding why very robust teachers produce qualitatively different output distributions.
- Javanmard and Soltanolkotabi, _"Precise Statistical Analysis of Classification Accuracies for Adversarial Training"_; Tanner et al., _"A High Dimensional Statistical Model for Adversarial Training: Geometry and Trade-Offs"_ , could inform why overconfident teachers arise and under what conditions one should expect TAS to concentrate.

On the question of when the same adversarial perturbation is effective for both the teacher and the student, two recent works are relevant:
- Balcan et al., _"Reliable learning in challenging environments"_ and Vilucchio et al., _"On the existence of consistent adversarial attacks in high-dimensional linear classification."_ Both consider settings where adversarial attacks must transfer across models; specifically, the second reference shows that for such attacks overparameterization (larger model capacity) may or may not help, which directly relates to the paper's finding that capacity gap alone does not explain distillation failures.

These would help situate the findings in a broader context and point toward more principled understanding of the phenomenon.

[Recommended] **Analyze what distinguishes ERTs from IRTs before distillation.** The ERT/IRT categorization is defined post hoc from distillation outcomes. For practitioners, the question is whether one can predict a teacher's effectiveness *before* running the student. Is there a relationship between the teacher's training method, calibration, or decision boundary geometry and its ERT/IRT status? Table 9 shows that some IRTs were trained with specific techniques — discussing whether any teacher-side properties are predictive would increase the paper's practical value.

[Minor] The expectation over $\mathcal{D}$ in Eq. 5 (the geometric mean defining $\bar{y}$) is only clarified in Algorithm 1 in the appendix. Since this is central to the decomposition in Eq. 6, the convention should be made explicit in the main text.

[Minor] Figure 1 is hard to parse due to overlapping points and abbreviated teacher names. Color-coding by TAS ratio in Figure 1a would make the message more immediately clear.

---

> ### Author Response · Authors · 2026-03-23
> **Reply to Reviewer 3qim**
>
> **1. Discuss the tightness and practical relevance of Lemma 2.**
>
> **A.** We agree with the reviewer that Lemma 2 can be loose in practice. Since the entropy on CIFAR-10 is upper-bounded by $\log(10)\approx 2.3$, we empirically observe that for some high-confidence cases (typical of IRTs), $\log m=\log(\min_i q_i)$ can fall below $-2.3$. We therefore clarify in the revision that the entropy proxy is empirically motivated rather than formally guaranteed by Lemma 2.
> In the revised manuscript, we have reframed our approach to explicitly state these mathematical limitations and present the entropy proxy as an empirically motivated heuristic rather than a strict formal justification. Furthermore, we have added a discussion regarding this gap in **Limitations in Section D**, leaving the establishment of a more generalized and formal theoretical framework for adversarial distillation to future research.
>
> **2. Strengthen the adversarial variance estimation or qualify its reliability.**
>
> **A.** To qualify the reliability of our adversarial variance (AVar) estimation, we conducted an extended sensitivity analysis on CIFAR-10 using a ResNet-18 student across four different teachers. We systematically varied the key estimation hyperparameters: the number of repetitions $K \in \{2, 3, 5\}$ and the number of splits $N \in \{2, 3, 4, 5\}$. This analysis has been added to **Section C.10 (Table 21)** of the revised manuscript.
>
> As presented in the table below, increasing the number of repetitions $K$ yields negligible differences in the estimated AVar. This confirms that our estimation is statistically stable and consistent across different random splits. Conversely, increasing the number of splits $N$ leads to an increase in the absolute AVar values. This is an inevitable consequence of reducing the subset size $|D_j^{(k)}|$ available for the adversarial training phase. Most importantly, the relative ordering of AVar among the four teachers remains consistent across all configurations of $K$ and $N$. This invariant ordering demonstrates that our estimation method reliably captures the comparative variance characteristics of different teachers.
>
>
> | Teacher | Default ($K=2, N=2$) | $K=3$ | $K=5$ | $N=3$ | $N=4$ | $N=5$ |
> | :--- | :---: | :---: | :---: | :---: | :---: | :---: |
> | Rebuffi2021Fixing | 0.0267 | 0.0273 | 0.0275 | 0.0441 | 0.0556 |0.0694 |
> | Chen2021LTD | 0.0059 | 0.0061 | 0.0060 |0.010 |0.0140  | 0.0168 |
> | Bartoldson2024Adversarial | 0.0834 | 0.0817 | 0.0805 | 0.1152 | 0.1405 | 0.1588 |
> | Gowal2021Improving | 0.3058 | 0.3016 | 0.3061 | 0.4495 | 0.5386 |0.6448 |
>
>
>
>
>
> **3. Engage more with the theoretical adversarial robustness literature.**
>
> **A.** We agree that situating our empirical findings within the broader theoretical literature is important. In the revised manuscript, we have expanded the **Discussion in Section D** to explicitly integrate the recommended theoretical frameworks. Specifically, we connect our observations to prior perspectives on geometry and accuracy. We also briefly note related theoretical views suggesting that robustness limitations can persist beyond capacity scaling. Finally, we reference the non-robust features framework as an interpretation of why some student-crafted perturbations may fail to transfer.
>
>
> **4. Analyze what distinguishes ERTs from IRTs before distillation**
>
> **A.** We agree that predicting a teacher's efficacy a priori is a highly practical question. While a definitive single metric remains elusive, our ongoing empirical analysis reveals two indicators. First, IRTs frequently employ massive diffusion-based data augmentation. We hypothesize this creates a highly complex decision boundary that capacity-constrained students struggle to match, directly leading to a scarcity of TAS. Second, our preliminary observations suggest that teachers whose feature representations align more closely with standard AT on students (e.g., PGD-AT or TRADES on student model) tend to act as ERTs. Building on these insights, a promising direction for future work is to leverage feature learning theory to formally establish this failure mechanism. Specifically, we hypothesize that highly robust IRTs enforce confident supervision on complex features the student cannot represent, compelling the student to memorize spurious noise. This induced noise memorization then provides vulnerable directions that attackers can easily exploit in test time, directly driving robust overfitting. We have added a discussion of these predictive properties and the failure mechanism to the **Future Work in Section D**.
>
>
>
>
> **5. Define geometric mean in text.**
>
> **A.** We have updated the text following Eq. 5 to define $\bar{\mathbf{y}}$ as the normalized geometric mean in the main text.
>
> **6. Figure 1 is hard to parse**
>
> **A.** We have updated Figure 1 by color-coding points by the TAS ratio as suggested.

---

### Review · Reviewer_MLHq · 2026-03-20

**Summary Of Contributions:**

This work studies the robust saturation effect. It proposes to categorizes robust teachers into two groups: effective robust teachers (ERTs) and ineffective robust teachers (IRTs). Then, with empirical results, it shows the characteristics of IRTs, such as overconfident predictions. By identifying the causes of the robust saturation effect, this work further proposes a sample-based weighting algorithm to up- or down-weight examples for adversarial distillation. The empirical results demonstrate the effectiveness of the proposed algorithm with various settings.

**Audience:**

Yes

**Audience Explanation:**

At least the results from the first part will interest researchers working on adversarial distillation, and may motivate some other algorithms.

**Claims And Evidence:**

Yes

**Claims Explanation:**

The main claim of this work has two part, the first part is about the hypothesis on the cause of the robust saturation effect; and the second part is about the effectiveness of the proposed algorithm.

The first part was supported by the experiments, illustrated in figure 1 and other relevant results in section 3. The second part was supported by empirical results in section 5.

**Requested Changes:**

There is no significant change from my perspective.

---

> ### Author Response · Authors · 2026-03-23
> **Reply to Reviewer MLHq**
>
> We thank the reviewer for evaluating our manuscript. We are glad that you found our empirical analysis of the robust saturation effect convincing, and our proposed sample-based weighting algorithm effective.

---

### Decision · Action_Editor_ASVu · 2026-04-29

**Recommendation:** Accept as is

**Additional Comments:**

In view of camera-ready, I encourage the authors to tone down the conclusions from C.3: I am not sure one can distinguish the effect of a stronger train-time attack from that of a larger TAS in that context.

**Audience:**

Yes

**Audience Explanation:**

All reviewers agree that the work is of interest to the TMLR community.

**Claims And Evidence:**

Yes

**Claims Explanation:**

All reviewers agree that the work's claims are adequately supported by experimental evidence. Figure 1 is a good example in this regard. In particular, the work shows that good transferability of adversarial examples is key to an empirically effective distillation process, and the proposed method (SAAD-C) appears to be delivering consistent improvements.